ecology/behaviour

diet, stable isotopes, movement, competition, partitioning, habitat selection

**Author for correspondence:**
Ryan R. Reisinger
e-mail: ryan.r.reisinger@gmail.com

# Foraging behaviour and habitat-use drives niche segregation in sibling seabird species

Ryan R. Reisinger[1,2,3], Tegan Carpenter-Kling[1,4], Maëlle Connan[1], Yves Cherel[2] and Pierre A. Pistorius[1,4]

[1]Marine Apex Predator Research Unit, Institute for Coastal and Marine Research and Department of Zoology, Nelson Mandela University, South Campus, Port Elizabeth 6031, South Africa
[2]Centre d'Etudes Biologiques de Chizé, UMR 7372 du CNRS-La Rochelle Université, 79360 Villiers-en-Bois, France
[3]Institute for Marine Sciences, University of California Santa Cruz, Santa Cruz, CA 95060, USA
[4]DST/NRF Centre of Excellence at the FitzPatrick Institute of African Ornithology, Department of Zoology, Nelson Mandela University, Port Elizabeth, South Africa

RRR, 0000-0002-8933-6875; TC-K, 0000-0001-8449-2409; MC, 0000-0002-1308-9118; YC, 0000-0001-9469-9489; PAP, 0000-0001-6561-7069

To mediate competition, similar sympatric species are assumed to use different resources, or the same but geographically separated resources. The two giant petrels (*Macronectes* spp.) are intriguing in that they are morphologically similar seabirds with overlapping diets and distributions. To better understand the mechanisms allowing their coexistence, we investigated intra- and interspecific niche segregation at Marion Island (Southern Indian Ocean), one of the few localities where they breed in sympatry. We used GPS tracks from 94 individuals and remote-sensed environmental data to quantify habitat use, combined with blood carbon and nitrogen stable isotope ratios from 90 individuals to characterize their foraging habitat and trophic ecology. Females of both species made distant at-sea foraging trips and fed at a similar trophic level. However, they used distinct pelagic habitats. By contrast, males of both species mainly foraged on or near land, resulting in significant sexual segregation, but high interspecific habitat and diet overlap. However, some males showed flexible behavioural strategies, also making distant, pelagic foraging trips. Using contemporaneous tracking, environmental and stable isotope data we provide a clear example of how sympatric sibling species can be segregated along different foraging behaviour dimensions.

# 1. Introduction

Competition among organisms for limited resources is central in shaping community structure and processes [1,2]. According to the competitive exclusion principle, competitors for identical resources cannot coexist [3]. It follows that resource use must be partitioned in some manner to alleviate similarity, and this segregation typically occurs along three axes: diet, space (habitat) or time [4,5]. Competitors may segregate along one axis while overlapping along others. Sympatric jaguar (*Panthera onca*) and puma (*Puma concolor*), for example, had highly overlapping space and time use, but segregated along the dietary axis, differing in the prey type and size they selected [6]. Alternatively, species may mitigate resource overlap by segregating habitat use. For example, a sibling pair of *Rhinolophus* bat species used different foraging habitat types when they occurred in sympatry, but this habitat preference disappeared in allopatry [7]. Even when species use the same resources and the same habitats, segregation along the temporal axis can allow species to exploit similar resources without direct competition. For example, a slight difference in the phenology of Adélie (*Pygoscelis adeliae*) and chinstrap (*P. antarcticus*) penguins means that the two species substantially reduced spatial overlap by foraging in similar areas a few weeks apart [8].

At the same time, males and females of the same species might be segregated on these same three axes, such that they have different diets (e.g. [9]), use different habitats (e.g. [10]) or time their activities differently (e.g. [11]). Two main hypotheses aim to explain these patterns in birds [12]. The 'social dominance hypothesis' posits that segregation results from the exclusion of subordinates by dominant conspecifics [12]. The 'specialization hypothesis' suggests that males and females segregate habitats and/or diets due to different requirements, opportunities and constraints related to their morphology, physiology and reproductive roles [12,13]. Competition, whether current or in species' evolutionary history, is frequently invoked as an explanation for niche segregation, but segregation may result from other factors including non-competitive evolutionary adaptation and stochastic processes [14].

Niche segregation among related species, and between the sexes, has often been found in seabirds (e.g. [8,15–20]) and two complementary tools have been particularly useful in providing this information. Tracking data allow us to quantify the space use of seabirds and, in conjunction with remote sensing data, allow us to quantify the environmental conditions—and thus the habitat—used by them [21]. Measurements of $\delta^{13}$C and $\delta^{15}$N stable isotope values in tissues such as blood and feathers are reliable biomarkers of foraging habitat and trophic level, respectively [22]. These complementary sources of information capture both scenopoetic (biophysical condition or setting) and bionomic (resource) axes of the species' niches (*sensu* [23]).

Two seabird species—the giant petrels (*Macronectes*)—are an intriguing example of morphologically similar species with overlapping diets and distributions, that display intra- and interspecific segregation [24–26]. Northern giant petrels *M. halli* (NGPs) and southern giant petrels *M. giganteus* (SGPs) were originally considered conspecific, but separate species were designated based on morphology and breeding phenology [27]. Their status as separate species has since been confirmed by genetic analyses with an estimated divergence *ca* 0.2 Ma [28,29]. Giant petrels breed at sites throughout the Southern Ocean, and breed in sympatry at a handful of sub-Antarctic islands [30]. Both species are sexually dimorphic, with males approximately 20–30% heavier than females [31]. While NGPs are slightly lighter than SGPs, the sexual dimorphism is more marked than interspecific size differences [25,31,32]. The parallel pattern in NGPs and SGPs led Hunter [32] and González-Solís *et al*. [31] to speculate that sexual segregation evolved before the two species diverged. Sexual dimorphism in NGPs, at least, is probably driven more by food competition avoidance rather than sexual selection [26,33,34].

Both male and female giant petrels feed on penguin and seal carrion during the breeding season [24]. Females, however, also include a large proportion of marine prey in their diet—cephalopods, fishes and crustaceans [24,35,36]. These diet differences are reflected in habitat use shown in tracking studies wherein females foraged in more distant offshore and pelagic environments, whereas males more often foraged onshore or inshore [31,33,37–41]. Spatial foraging segregation between the sexes thus appears to limit intraspecific competition [31,38]. Niches may be segregated along a time axis in addition to the spatial and dietary axes, and breeding phenology is thus an additional mechanism through which competition can be reduced or avoided (e.g. [42]). NGPs commence breeding approximately five to six weeks before SGPs, and this may be one of the most important reproductive isolating mechanisms between the species [25,32,41,43]. This allows the two species to use different seasonal resource pulses near breeding colonies. At Marion Island, for example, NGPs scavenge

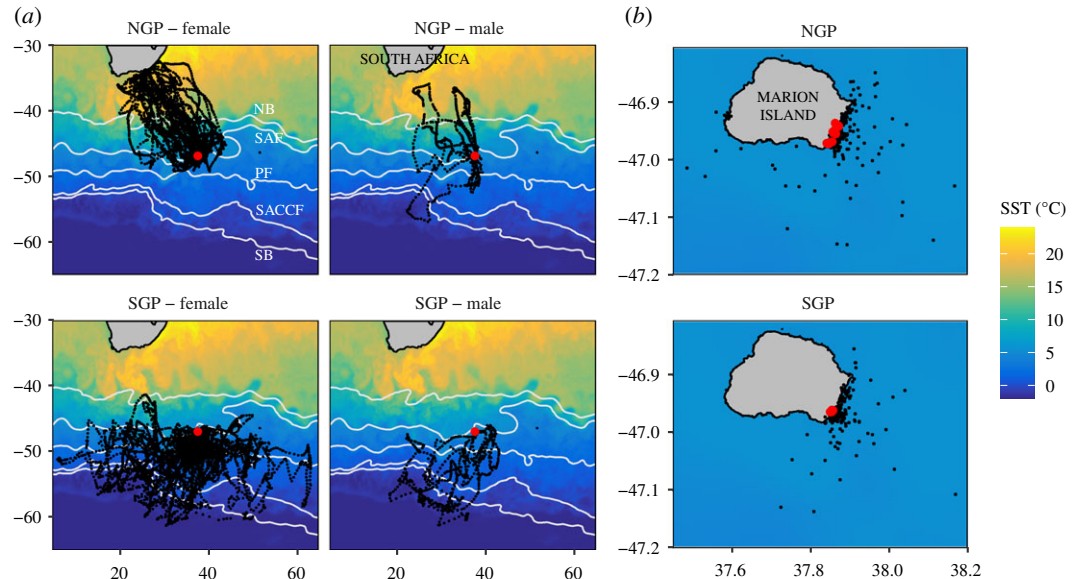

**Figure 1.** Locations of 49 northern giant petrels (upper panels) and 45 southern giant petrels (lower panels). (*a*) All locations, with Marion Island indicated with red points. (*b*) Focuses on Marion Island and shows only locations from short distance (less than 50 km) movements of 25 males and two females with nest locations indicated with red points. The background image in (*a*) and (*b*) shows sea surface temperature (SST; °C) for 1 October 2015. Sea surface temperature was the primary variable distinguishing habitat use among species and sexes (figure 3). White lines in (*a*) show general locations of, from north to south, the northern boundary of the Antarctic Circumpolar Current (NB) that corresponds with the Subtropical Front, the Sub-Antarctic Front (SAF), the Antarctic Polar Front (PF), the Southern Antarctic Circumpolar Current Front (SACCF) and the southern boundary of the Antarctic Circumpolar Current (SB) [44].

elephant seals and fur seals which breed slightly earlier in summer than the penguins mainly scavenged or predated by SGPs [43]. However, there is still substantial overlap in the breeding period of the two species, when they are foraging under a central place constraint.

Given the close evolutionary relationship between these two species, their highly similar morphology, sympatric breeding at several sites for substantial periods of time and similar diets, there intuitively seems to be strong potential for competitive exclusion between these species. The giant petrels therefore allow insight into the mechanisms of competition avoidance or niche segregation between species. This has stimulated the many studies of the ecology of giant petrels, and they have become a well-known example of ecological segregation (e.g. [26]). We build on this body of work by integrating stable isotope analysis, tracking data and remote-sensed environmental data for sympatric-breeding *Macronectes* populations studied over three consecutive breeding seasons. We do this to test the hypothesis that there is niche segregation in sympatric populations of giant petrels at Marion Island, Southern Indian Ocean, a site where the behaviour of these species is poorly known. Given the species' recent divergence and marked sexual dimorphism, we predict that segregation between sexes should be greater than that between species, but that the two species should also show some segregation along one or more niche-axes, which has developed in the 0.2 Myr since their divergence [29]. To test this hypothesis, we use tracks from animal-borne GPS devices, and the carbon and nitrogen stable isotope ratios ($\delta^{13}$C and $\delta^{15}$N) in the blood plasma of the tracked, and some untracked, individuals.

# 2. Methods

## 2.1. Study site

The Prince Edward Islands (46.9° S, 37.7° E), comprising Marion Island and Prince Edward Island, are located on a shallow rise surrounded by water approximately 5000 m deep. There is high mesoscale oceanographic variability in the vicinity of the islands, stemming from the frequent presence of eddies spawned as the west-flowing Antarctic Circumpolar Current interacts with the Southwest Indian Ridge upstream of the islands. The islands are situated in the Polar Frontal Zone, which is delineated by the Antarctic Polar Front in the south and the Sub-Antarctic Front in the north (figure 1)

[45]—areas attractive to various marine predators [46]. The biophysical characteristics of the surrounding ocean thus show strong latitudinal patterns. This includes a latitudinal gradient in the $\delta^{13}C$ value of particulate organic matter [47,48], which has been exploited to infer animal movements in the Southern Ocean [49,50]. In 2013/2014, Marion Island had approximately 443 breeding pairs of NGPs (increasing since the early 2000s) and approximately 1583 breeding pairs of SGPs (stable since the late 1990s) [51,52]. Together with Prince Edward Island, this represents around 6% of their respective global populations [51]. Millions of seabirds and seals breed at these islands in summer [53].

## 2.2. Fieldwork

Species were differentiated by coloration of the bill tip [32] with nesting site and breeding phenology providing additional context for species assignment [43]. We deployed GPS loggers (CatLog-S, Perthold Engineering LLC, USA, $50 \times 22 \times 8$ mm, 34 g) on 120 individuals (20 individuals per species per year) near Kildalkey Bay on the southeastern side of Marion Island, during their incubation period in late September and early October in 2015, 2016 and 2017 (electronic supplementary material, table S1). The tracking periods for the two species were mostly overlapping (electronic supplementary material, figure S1). The loggers were attached to body feathers between the scapula using four to five strips of waterproof adhesive TESA® tape (Beiersdorf AG, GmbH, Hamburg, Germany), and secured with cable ties and cyanoacrylate glue (Loctite 401®). We estimated handling time was approximately 5–10 min per bird. We retrieved 96 loggers with data (50 NGPs and 46 SGPs).

At retrieval, we measured the birds' culmen length and bill depth (gonys depth). Culmen lengths of males and females are non-overlapping [32,34]. We found two clearly distinct groups based on culmen length (see Results and electronic supplementary material, figure S2) and thus putatively assigned individuals with culmen lengths greater than 97 mm as males, and those with culmen lengths less than 93 mm as females. For four birds with no bill measurements, sex was determined genetically using the primers 2550F and 2718R (adapted from [54]). We additionally determined the sex of 22 other birds genetically to confirm that sex assignments based on bill measurements were generally valid.

Upon retrieval of GPS data loggers, usually around two weeks after deployment (electronic supplementary material, table S1), approximately 1 ml of blood was collected from the tarsus vein using a slightly heparinized syringe with a 25 gauge needle. Approximately half of the blood sample was stored directly in 70% ethanol. The other half was centrifuged within 3–4 h after collection, separated into red blood cells and plasma, and stored in 70% ethanol. Storage in 70% ethanol was preferred to drying because of the field conditions and it does not significantly affect $\delta^{13}C$ and $\delta^{15}N$ values [55]. Samples were then stored at −20°C as soon as possible until preparation for stable isotope analysis. Blood was sampled from 105 individuals, including 82 of the birds successfully GPS-tracked.

## 2.3. GPS analyses

Data analyses were done in the R environment [56]. GPS loggers recorded locations at approximately 60 min intervals. Tracks were filtered based on a $30 \text{ m s}^{-1}$ speed threshold [33,41] in the *argosfilter* package [57]. For visualization and broad description, foraging 'trips' were identified based on displacement from the nest. These trips were delineated by 3 or more hours of successive locations further than 200 m from the nest location. After filtering (which removed approx. 0.02% of locations) and visual inspection, the tracks of 94 individuals (49 NGPs and 45 SGPs) were retained for further analyses. Data exploration indicated two distinct movement strategies—nearby and distant trips. We quantitatively distinguished these two strategies by calculating quantile breaks on trip distance using the *classInt* package [58].

To quantify spatial overlap between species and sexes, we first calculated utilization distributions, using the *adehabitatHR* package [59]. Bandwidth (*h*) selection by least-squares cross validation failed to converge and we thus selected *h* using the 'ad-hoc' method, whereby an initial, high value for *h* is chosen and incrementally decreased until the resulting utilization distribution starts to break up (undersmoothing) [60]. Values were $h = 0.8°$ for all trips together and $h = 0.005°$ for nearby trips only. Before calculating utilization distributions, we excluded locations where the preceding displacement required a speed greater than $7.2 \text{ m s}^{-1}$, with the rationale that these locations were unlikely to represent foraging and would thus bias the utilization distributions. The speed threshold was calculated using the Fisher–Jenks algorithm in the *classInt* package [58]. We calculated 95% utilization distributions to represent most of the foraging range of each group, and 50% utilization distributions to represent core areas. We then calculated an overlap measure—Bhattacharyya's affinity [61]—among

groups, also using *adehabitatHR*. We tested the null hypothesis of no spatial segregation by permuting the individual track labels (sex or species, as appropriate) 1000 times and calculating overlap for each permutation (e.g. [15]). Since interannual variation in resource availability and environmental conditions may influence foraging behaviour, we constrained the permutations by year. However, for other habitat analyses (including the random forest model described below) we pooled the data for different years. The *p*-values for the permutation tests were estimated as the proportion of times the observed overlap was greater than the permuted overlap.

To characterize the at-sea habitat used by individuals, we collated seven candidate environmental variables at each GPS location using the *raster* [62], *raadtools* [63] and *xtractomatic* [64] packages. These were: sea surface temperature, chlorophyll-a concentration, ocean depth, sea surface height anomaly, meridional wind velocity, zonal wind velocity and eddy kinetic energy (details in electronic supplementary material, table S2). Dynamic variables (except for chlorophyll-a concentration) were obtained at a daily resolution and matched to the date of each location. Chlorophyll-a concentration was matched monthly to lower the amount of missing data due to cloud cover. We then related these variables to species and sex using a random forest classification model—a fast and accurate method that can classify multiple target classes [65]. The fitted random forest model predicts, for each at-sea GPS location, to which group (NGP or SGP, male or female) the location belongs based on the environmental covariates. We fitted the model in the *randomForest* R package [66], growing 1000 trees. We assessed variable importance as the mean decrease in Gini index. To avoid collinearity among variables, we tested for pairwise correlation among variables. No variables had Spearman's $R > |0.70|$ [67], so we retained all variables in the models.

## 2.4. Stable isotope analyses

Stable isotope ratio values in blood plasma integrate information over the short term (days) [68,69]. We focused on blood plasma here since the integration time should correspond with the tracking period (days to weeks). Blood plasma was dried at 50°C for 24–48 h before being powdered. High lipid content in tissues causes lower $\delta^{13}C$ values because these lipids are depleted in $^{13}C$, biasing inference. This can be detected by high C : N ratios and lipid removal is advised for tissues with a C : N ratio greater than 3.5 for aquatic animals [70]. However, lipid removal unpredictably changes $\delta^{15}N$ values [71]. Thus, where possible, we divided each plasma sample into two fractions: lipids were extracted from one half while the other half of the sample was sent directly for stable isotope analysis without lipid extraction. Lipids were removed by immersing powdered plasma in a 2 : 1 chloroform : methanol solution with a solvent volume three to five times greater than sample volume. Samples were then vortexed for 10 s every 10 min for 1 h before being centrifuged for 5 min. The supernatant containing lipids was discarded, and samples dried at 50°C overnight.

Sample aliquots (approx. 0.4 mg) were analysed for carbon and nitrogen stable isotope ratios by combusting them in a Flash 2000 organic elemental analyser and passing gases through a Delta V Plus isotope ratio mass spectrometer via a Conflo IV gas control unit (Thermo Scientific, Germany). All samples were processed at the Stable Light Isotope Unit at the University of Cape Town, South Africa. Replicate measurements of internal laboratory standards indicated minimal standard deviations (Merck gel: $\delta^{13}C = 0.2‰$, $\delta^{15}N < 0.1‰$; valine: $\delta^{13}C < 0.2‰$, $\delta^{15}N = 0.1‰$; seal bone: $\delta^{13}C < 0.2‰$, $\delta^{15}N < 0.1‰$). All internal standards were calibrated against International Atomic Energy Agency standards. Carbon is expressed in terms of its value relative to Vienna PeeDee Belemnite, while nitrogen is expressed in terms of its value relative to atmospheric nitrogen. Hereafter, we use $\delta^{15}N$ values from the raw plasma samples and $\delta^{13}C$ values from the lipid-extracted plasma samples.

For further analyses, we used $\delta^{13}C$ and $\delta^{15}N$ values for the 90 birds where there was enough plasma for lipid extraction; 75 of these birds were tracked. We assessed multivariate normality with Mardia's skewness and kurtosis coefficients [72], tested in the *MVN* package [73]. Differences in stable isotope ratio values among groups (species × sex) were tested using a multivariate analysis of variance (MANOVA), followed by pairwise MANOVAs. To characterize the isotopic niche of each group, we calculated standard ellipse areas corrected for small sample sizes (SEAc), in the *SIBER* package [74]. As a measure of isotopic niche overlap, we calculated SEAc overlap as a proportion of the sum of the non-overlapping SEAc areas. We tested these overlap values against a null distribution using the permutation procedure described above for the tracking data. Again, however, we pooled data from the different years for the general stable isotope analyses.

While $\delta^{15}N$ values are mainly used to indicate the trophic position of animals, they are influenced by differences in baseline $\delta^{15}N$ values that reflect the isotopic gradients in the Southern Ocean (e.g. [49,75]). These gradients are captured mainly in the $\delta^{13}C$ values, but when comparing $\delta^{15}N$ values originating

from different ecosystems, differences in baseline $\delta^{15}N$ values need to be considered. To account for this effect when looking at trophic position, we fitted a linear regression of $\delta^{15}N$ values against $\delta^{13}C$ values and calculated the Studentized residuals from this regression (the $\delta^{15}N$ Studentized residuals) [76], giving us the relative trophic positions of individuals while controlling for the geographical source of their diet (the $\delta^{13}C$ values), as far as possible.

# 3. Results

## 3.1. Bill measurements

Molecular sexing confirmed all putative sex assignments based on bill measurements. Culmen length of putative males ranged from 98.2 to 110.3 mm, and from 87.9 to 95.7 mm in putative females. There was no clear differentiation in bill depth, which ranged from 27.8 to 45.0 mm. Neither culmen length ($t_{84.15} = 0.628$, $p = 0.532$) nor bill depth ($t_{83.33} = -0.182$, $p = 0.856$) was significantly different between species, indicating greater intersexual size differences than interspecific differences (electronic supplementary material, figure S2).

## 3.2. GPS tracking

Birds were tracked for 7.8–31.0 days (mean = 16.6 days). After trimming the tracks to exclude locations on the nest, tracks were 1.4–24.3 days long, with a mean of 9.1 days (electronic supplementary material, figure S1). Birds showed two principal foraging strategies: one group (24 individuals, all males [10 NGPs, 14 SGPs]) made nearby trips only (maximum distance from nest = 8–50 km) (figure 1; electronic supplementary material, table S1). These short trips were mainly forays to seal and penguin rookeries and inshore waters around the island, often interspersed with brief returns to the nest (electronic supplementary material, figure S3). The second group (70 individuals: 59 females [31 NGPs, 28 SGPs] and 11 males [8 NGPs, 3 SGPs]) made one or two clearly identifiable long-distance trips to sea (maximum distance from nest = 69–2344 km) and some nearby trips (electronic supplementary material, figure S3 and table S1). Thus, females of both species made only distant trips while males showed both behaviours—making distant trips or remaining near the nest (electronic supplementary material, table S1). Individuals that made only nearby trips had 26.0–92.3% of their locations on land (mean = 50.3%). Among the 70 individuals that made distant trips, trip duration was similar for males (mean = 10.4 days) and females (mean = 10.2 days; $t_{15.26} = 0.175$, $p = 0.863$). The culmen lengths of males that made long trips were not statistically different from those of males that did not make long trips ($t_{21.57} = -0.520$, $p = 0.609$).

There was a marked contrast in the space use of the two species at sea (figures 1 and 2). NGPs travelled mainly northwest, often to the South African shelf and shelf edge (figure 1). Locations were thus predominantly at latitudes north of Marion Island (mean = 43.50° S; figure 3). In sharp contrast, SGPs travelled mainly south, east and west (figure 1), with positions almost entirely south (mean = 50.55° S) of the island's latitude (figure 3). This segregation reflects the very different habitats used by each species and sex (figure 3). In the random forest model, the most important environmental variables in the classification of the four groups were sea surface temperature (mean decrease in Gini index = 2876), chlorophyll-a concentration (1387) and depth (1090). The remaining variables were substantially less important (753–864). The top three variables show how NGPs use the comparatively warm, productive waters north of the Subtropical Front with many locations in shelf waters (figure 3). SGPs mainly used colder, pelagic waters south of the Sub-Antarctic Front and Antarctic Polar Front and did not encounter the relatively high productivity areas that NGPs did (figure 3). The random forest model had an overall out-of-bag error rate of 10.4%. However, the model could more accurately classify females (error rate = 5.3% for NGP females and 4.3% for SGP females) than males (error rate = 34.3% for NGP males and 41.7% for SGP males).

Patterns of overlap were the same for the overall (95%) and core (50%) utilization distributions. Female NGPs and SGPs showed significantly lower overlap than expected (table 1). However, male NGPs and SGPs were not significantly segregated. Male and female NGPs showed significantly lower overlap than expected, as did male and female SGPs (table 1). These features are evident in plots of the groups' utilization distributions (figure 2).

## 3.3. Stable isotope analyses

Stable isotope ratio values differed significantly among groups (Wilks' lambda = 0.179, $p < 0.001$). Specifically, there were significant differences between all groups tested (table 2). NGP females had the highest $\delta^{13}C$ values

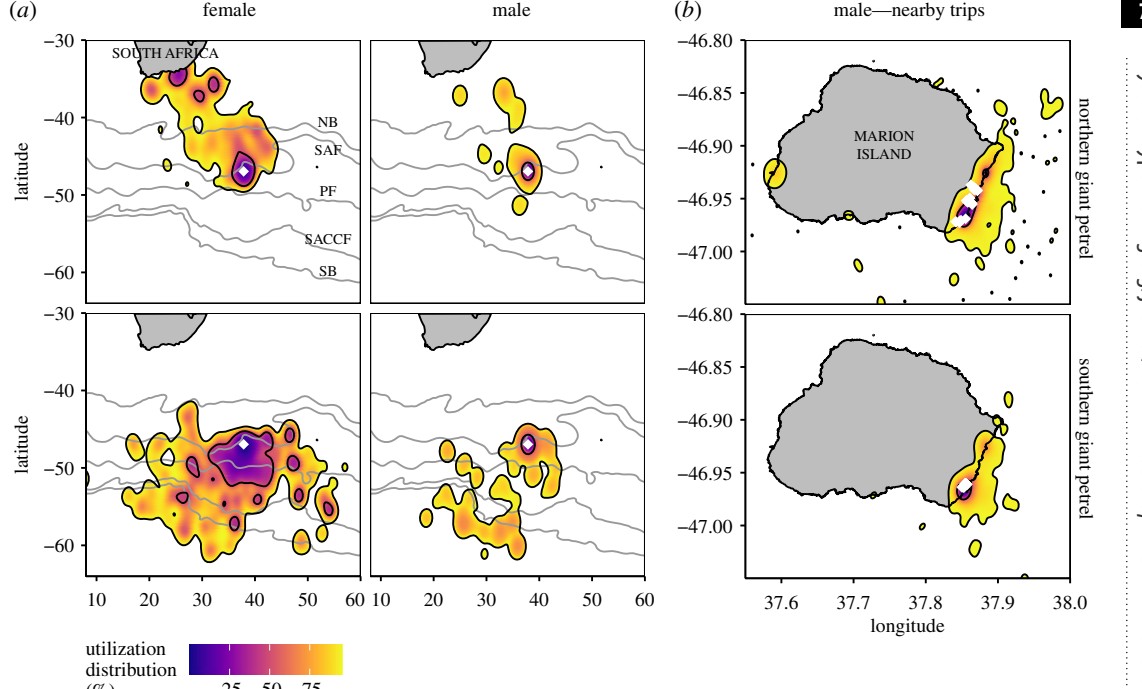

**Figure 2.** Kernel utilization distributions (up to 95%, with black contours at 50% and 95%) showing spatial overlap and segregation among (*a*) all 94 giant petrels tracked from Marion Island (white diamond) and (*b*) 41 giant petrel males that made short trips (less than 50 km) from their nests (white diamonds). Grey lines in (*a*) show general locations of oceanographic fronts, as in figure 1. Overlap values (and significant segregation among groups) are shown in table 2.

and $\delta^{15}N$ values. NGP males had the next-highest $\delta^{13}C$ values and $\delta^{15}N$ values. SGP females had the lowest $\delta^{13}C$ values and second-lowest $\delta^{15}N$ values. SGP males had the lowest $\delta^{15}N$ values (table 3 and figure 4). NGP males had the largest isotopic niche, followed by NGP females, SGP females and SGP males (table 3 and figure 4). SEAc overlap was significantly lower than expected between all groups tested (table 4). In the 20 individuals that made only short trips (less than 50 km), values were significantly different between SGPs and NGPs (Wilks' lambda = 0.583, $p = 0.001$). SGPs had lower $\delta^{13}C$ values (SGP = $-22.0 \pm 0.4‰$, NGP = $-20.8 \pm 1.0‰$) and slightly lower $\delta^{15}N$ values (SGP = $11.8 \pm 0.3‰$, NGP = $12.2 \pm 0.5‰$).

In the 55 individuals that made long trips, values were significantly different between SGPs and NGPs (Wilks' lambda = 0.306, $p < 0.001$). SGPs had much lower $\delta^{13}C$ values than NGPs (SGP = $-22.7 \pm 0.5‰$, NGPs = $-20.0 \pm 1.1‰$) and much lower $\delta^{15}N$ values (SGP = $12.7 \pm 0.4‰$, NGPs = $14.3 \pm 0.8‰$). However, in these individuals, mean latitude values were strongly correlated with $\delta^{13}C$ values (Pearson's R = 0.83) and correlated with $\delta^{15}N$ values (Pearson's R = 0.74). After regressing $\delta^{15}N$ values against $\delta^{13}C$ values for all individuals, males had lower values than females (figure 5*a*). However, among at-sea foragers only, the $\delta^{15}N$ Studentized residuals were similar among groups (figure 5*b*).

# 4. Discussion

Through the combined use of GPS tracking and stable isotope analyses, we demonstrate significant inter- as well as intraspecific niche segregation in giant petrels breeding in sympatry at Marion Island. During incubation, segregation occurred along at least two axes—isotopically distinct food resources (diet) and differential habitat use (space)—illustrating how environmental resources may be partitioned among similar animals. While several studies have shown facets of intra- and interspecific segregation in giant petrels, here we show this segregation along multiple axes, by integrating contemporaneously collected tracking, environmental and stable isotope data.

## 4.1. Overall segregation patterns

Within species, males and females typically used isotopically distinct resources and different habitats. Males of both species tended to stay close to the island where they were not spatially segregated,

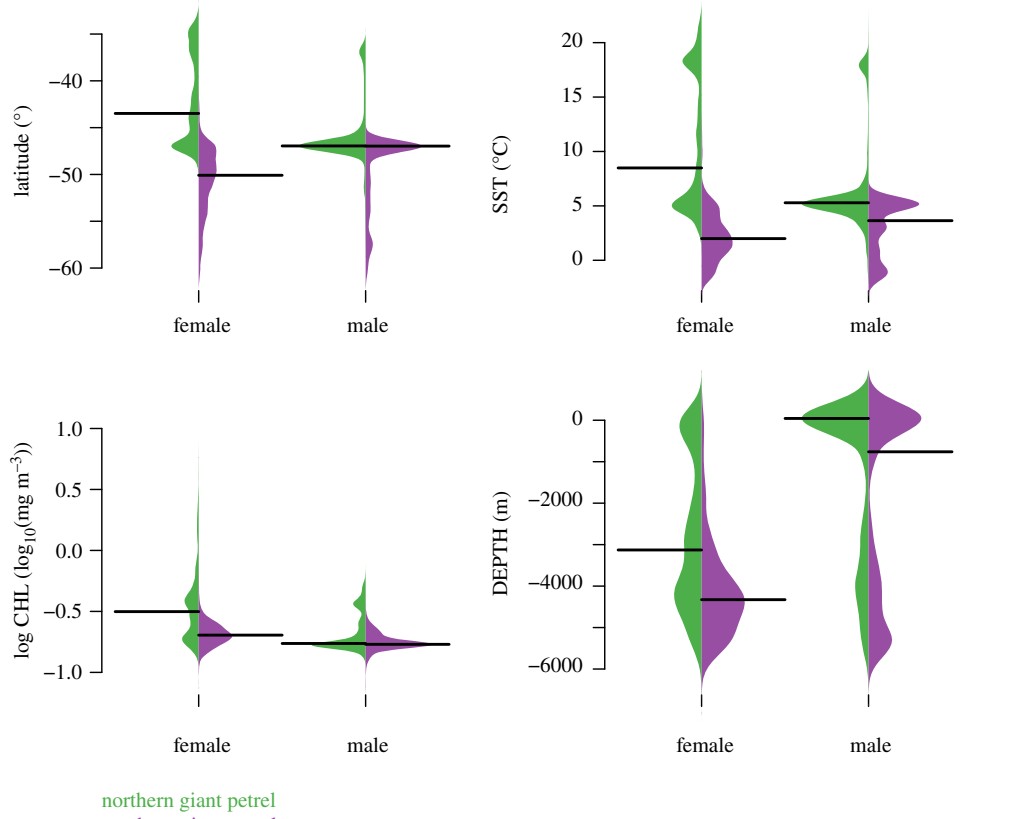

northern giant petrel
southern giant petrel

**Figure 3.** Density distributions of environmental conditions along the tracks of giant petrels tracked from Marion Island. Latitude (°) is shown as well as the three most important variables in a random forest model predicting group membership (sex and species). In order of importance: sea surface temperature (SST,℃), $\log_{10}$ of chlorophyll-a concentration (CHL, mg m$^{-3}$) and depth (DEPTH, m). Black lines show the mean of each distribution.

**Table 1.** Kernel utilization distribution overlap (Bhattacharyya's affinity) among giant petrel species and sexes. Significant segregation (at $\alpha = 0.05$, in bold) was identified by comparing the real (observed) overlap to a distribution of overlaps from 1000 permutations of the track labels, constrained by year. Overlaps of the (a) 95% and (b) 50% utilization distributions (UD) are shown. NGP, northern giant petrel; SGP, southern giant petrel.

| overlap between | observed overlap | permuted overlap (mean ± s.d.) | *p*-value | *p*-value 95% CI |
|---|---|---|---|---|
| (a) 95% UD | | | | |
| male NGPs and male SGPs | 0.76 | 0.79 ± 0.02 | 0.921 | 0.902–0.937 |
| **female NGPs and female SGPs** | **0.34** | **0.74 ± 0.02** | **0.000** | **0.000–0.004** |
| **male NGPs and female NGPs** | **0.65** | **0.81 ± 0.02** | **0.000** | **0.000–0.004** |
| **male SGPs and female SGPs** | **0.61** | **0.74 ± 0.03** | **0.001** | **0.000–0.006** |
| (b) 50% UD | | | | |
| male NGPs and male SGPs | 0.47 | 0.46 ± 0.03 | 0.353 | 0.323–0.384 |
| **female NGPs and female SGPs** | **0.24** | **0.37 ± 0.02** | **0.000** | **0.000–0.004** |
| **male NGPs and female NGPs** | **0.33** | **0.45 ± 0.03** | **0.001** | **0.000–0.006** |
| **male SGPs and female SGPs** | **0.22** | **0.39 ± 0.02** | **0.000** | **0.000–0.004** |

exploiting foraging areas on land near their nests. However, a few males showed similar foraging strategies to females, making distant at-sea foraging trips and exploiting the same resources and habitats as females. Males of the two species exploited resources at a similar trophic level ($\delta^{15}$N Studentized residuals), but with different origins, evidenced by slight differences in $\delta^{13}$C values. Thus,

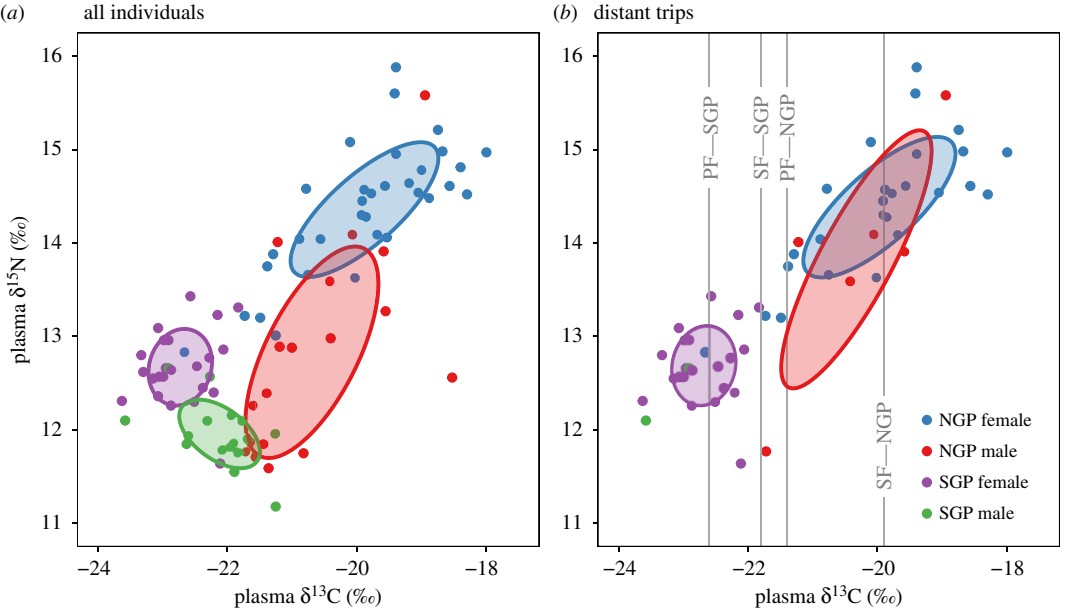

**Figure 4.** Biplot showing blood plasma stable isotope ratio values (filled points) of (*a*) all 90 northern (NGP) and southern (SGP) giant petrel males and females. (*b*) 55 giant petrels that made distant foraging trips. Filled ellipses show the small sample size corrected standard ellipse area (SEAc) for each group. For (*b*), only two SGP males (green points) made distant trips, and their SEAc could thus not be calculated. Vertical lines in (*b*) show the species-specific $\delta^{13}$C values corresponding approximately with the Antarctic Polar Front (PF) and Subtropical Front (SF) [75].

**Table 2.** Pairwise multivariate analysis of variance (MANOVA) for differences in $\delta^{13}$C and $\delta^{15}$N values between groups of northern (NGP) and southern (SGP) giant petrel males and females. All results were significant (at $\alpha = 0.05$).

| comparison | Wilks' lambda | *p*-value |
|---|---|---|
| male NGPs and male SGPs | 0.552 | <0.001 |
| female NGPs and female SGPs | 0.270 | <0.001 |
| male NGPs and female NGPs | 0.539 | <0.001 |
| male SGPs and female SGPs | 0.475 | <0.001 |

**Table 3.** Mean blood plasma $\delta^{13}$C and $\delta^{15}$N values in the blood plasma of 90 giant petrels and size of the isotopic niche of each group, measured as the small sample size corrected standard ellipse area (SEAc).

| species | sex | $\delta^{13}$C (‰) | | $\delta^{15}$N (‰) | | SEAc (‰$^2$) |
|---|---|---|---|---|---|---|
| | | mean | s.d. | mean | s.d. | |
| NGP | female | −19.9 | 1.1 | 14.4 | 0.7 | 1.6 |
| NGP | male | −20.7 | 1.0 | 12.8 | 1.1 | 2.7 |
| SGP | female | −22.7 | 0.5 | 12.7 | 0.4 | 0.6 |
| SGP | male | −22.1 | 0.6 | 12.0 | 0.4 | 0.6 |

males of the two species were isotopically segregated (along the $\delta^{13}$C axis) despite using the same foraging habitats. Females of the two species used food resources at a similar trophic level but segregated spatially and thus used different habitats (different water masses) and thus different prey.

## 4.2. Interspecific segregation in females

Females of both species made distant at-sea foraging trips but used distinct pelagic habitats: female SGPs used colder, deeper habitats in the Sub-Antarctic and Antarctic Zones, while female NGPs used

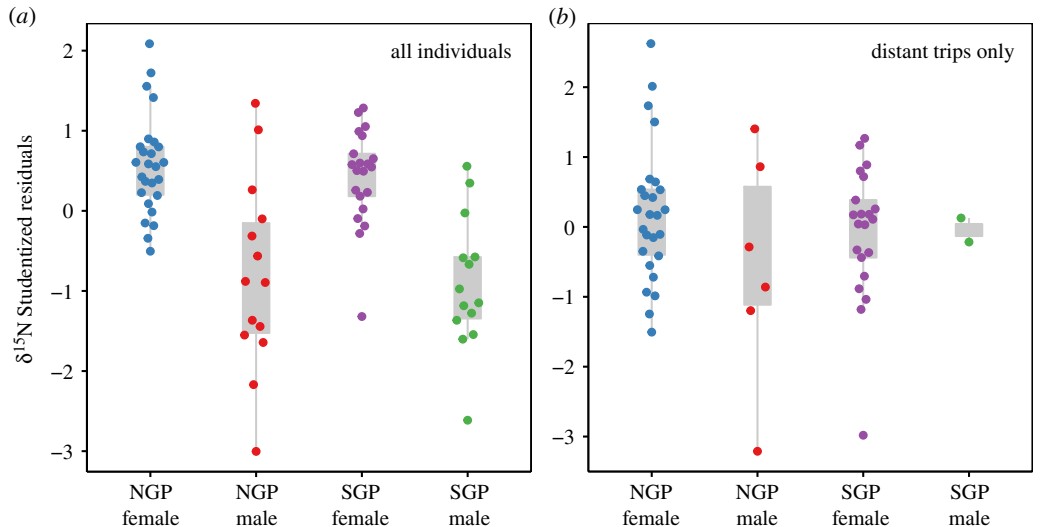

**Figure 5.** Plasma $\delta^{15}$N Studentized residuals for (*a*) all 75 giant petrels that were tracked and sampled; (*b*) 55 giant petrels that were sampled and made distant (greater than 50 km) foraging trips. The residuals are from a regression of $\delta^{15}$N against $\delta^{13}$C values, to correct for the strong latitudinal isoscape in the Southern Ocean, which influences $\delta^{15}$N values, but is primarily represented in $\delta^{13}$C values (figure 4). SGP, southern giant petrel; NGP, northern giant petrel.

**Table 4.** Isotopic niche (small sample corrected standard ellipse area; SEAc) overlap among giant petrel species and sexes. SEAc overlap between two groups is expressed as a proportion of the sum of the non-overlapping SEAc areas of the two groups. Significant isotopic niche segregation (at $\alpha = 0.05$) was identified by comparing the real (observed) overlap to a distribution from 1000 permutations of the individual labels, constrained by year.

| SEAc overlap between | observed SEAc overlap | permuted SEAc overlap (mean ± s.d.) | *p*-value | *p*-value 95% CI |
|---|---|---|---|---|
| male NGPs and male SGPs | 0.008 | 0.445 ± 0.124 | 0.000 | 0.000–0.004 |
| female NGPs and female SGPs | 0.000 | 0.582 ± 0.121 | 0.000 | 0.000–0.004 |
| male NGPs and female NGPs | 0.246 | 0.568 ± 0.127 | 0.000 | 0.000–0.004 |
| male SGPs and female SGPs | 0.000 | 0.452 ± 0.145 | 0.000 | 0.000–0.004 |

comparatively warm subtropical habitats, frequently along the continental shelf-break of South Africa. In these habitats (after accounting for isotopic baseline differences using the $\delta^{15}$N Studentized residuals), they fed at a similar trophic level, probably on predatory cephalopods and fishes. Females of the two species thus shared a similar trophic level. However, they segregated spatially—foraging in different pelagic habitats and exploiting the different prey that occur in these water masses—and they were thus isotopically segregated. Female NGPs and SGPs tracked during incubation at Bird Island, South Georgia, had low spatial overlap (kernel utilization distribution overlap = 0.167) but they used only slightly different at-sea habitat [41]. The segregation was not as extreme as that which we observed, and Granroth-Wilding & Phillips [41] suggested that allochrony through slightly different breeding times was a primary segregation mechanism between giant petrel species at South Georgia. Such differences among populations illustrate the likely role of local conditions, such as oceanography and prey distribution and availability, in influencing the foraging strategies of giant petrels. The specific situation of Marion Island near the Antarctic, sub-Antarctic and subtropical waters, allows giant petrels at Marion Island to exploit these very different habitats and corresponding different prey types.

## 4.3. Interspecific segregation in males

Males, in contrast to females, typically foraged on land or inshore near their breeding sites, resulting in significant sexual segregation, but high interspecific habitat overlap. Aggressive competition for carrion is commonly seen between male SGPs and NGPs, and it is possible that the dominance of some males excludes others from carrion resources, forcing them to forage at sea or at lower quality sites on land [77].

However, culmen lengths (a proxy of body size) of males that made long trips were not different from those of males that did not make long trips. Future work could investigate resource-use patterns in males, with respect to year-to-year variation in carrion availability.

While the trophic level of males that foraged on or near land was similar between species, the slightly lower $\delta^{13}$C values of SGP males suggest that they use a different prey resource on land, with a more southerly origin (since $\delta^{13}$C baseline values in the Southern Ocean decrease at more southerly latitudes [48]), leading to subtle isotopic segregation. If males frequently alternate their foraging strategy between scavenging on land and pelagic trips, this might influence their $\delta^{13}$C values, explaining the small difference, but this is not evident in the $\delta^{15}$N values or supported by the short integration time of blood plasma, which corresponds well with the tracking periods. The five to six week difference in breeding timing between the species has been suggested to be an important segregation mechanism between the two species in general [25,41,43], but it does not free males from broad spatial overlap, as we show with our data collected near-simultaneously for both species.

## 4.4. Intraspecific segregation

The $\delta^{15}$N Studentized residual values suggest that males feed at a lower trophic level than females. This is unexpected, as females feed on pelagic prey such as fishes and cephalopods while carrion dominates the diet of males [24,32,35]; however, this was also found for SGPs breeding in Antarctica and Patagonia [78]. This suggests that during the tracking period the diet of males was dominated by species feeding at a relatively low trophic level. Rather than feeding on seals, which are mainly piscivorous at Marion Island [79], the low $\delta^{15}$N values of male giant petrels suggest they feed on crustacean-feeders [80,81]. However, a better isotopic characterization of the potential prey field is required to resolve the potential diet composition of giant petrels at Marion Island.

Several males used similar foraging strategies to females, resulting in some niche overlap between the sexes. Males that made distant foraging trips fed on similar trophic-level prey (indicated by their higher $\delta^{15}$N Studentized residual values) to females. Different strategies, especially among NGP males, also resulted in the largest isotopic niche. This flexibility in foraging strategies has been observed in NGPs and SGPs at South Georgia, where both males and females showed plastic foraging behaviour [41], although females at South Georgia [36,41] and Patagonia [36] displayed more foraging flexibility than we observed. Our results were more similar to the consistency reported for females at South Georgia in a different study [38], highlighting the possibility of significant interannual variation and the utility of integrating stable isotope and tracking data. While our results reinforce the broad pattern of males having a more coastal distribution while females forage in pelagic environments [31,33,37–40,82], we show that males can be very flexible in their foraging strategies. Females may be competitively excluded from carrion resources ashore and forced more often to forage at sea due to their smaller body size, while males can supplement their on-land carrion foraging with at-sea foraging when necessary. In winter, for example, studies elsewhere show that both sexes forage at sea, when carrion availability on land is low [38,41,83]. Additionally, females may have higher energy requirements following egg-laying, or require specific nutrients, which would influence their foraging behaviour [84,85].

## 5. Conclusion

We integrated fine-scale tracking and stable isotope analyses in sympatric giant petrels, showing the extent and mechanism of segregation within and between these sibling species along two axes. Together with earlier work, our results show how females and males forage on different prey in different habitats, and how NGP and SGP females forage in different areas at sea. However, the specific patterns are tied to local conditions, necessitating some flexibility in foraging strategies that is evident among giant petrel populations studied thus far. The intraspecific segregation between males and females, which may be driven by competitive exclusion of the smaller females from carrion resources ashore, is more marked than interspecific segregation. However, the geographical segregation of northern and southern giant petrels at sea is striking. As suggested by Hunter [32], the parallel pattern of sexual segregation in the two species indicates that sexual segregation arose before speciation in the giant petrels 0.2 Ma [29], and the use of different geographical foraging areas at sea may represent the foraging preferences of ancestral populations that have persisted after secondary contact [32]. These foraging preferences may have arisen in sympatry, not necessarily due to competition, or evolved in allopatry. Variation in foraging patterns among giant petrels breeding at

Marion Island exposes species and sexes to different threats, which should be investigated in more detail (e.g. [40,86]) and potentially considered in their conservation and management. Future work should also address the role of interannual variation in resource availability on segregation patterns, particularly in males ashore.

Ethics. Ethics approval was granted by the Nelson Mandela University Research Ethics Committee (A14-SCI-ZOO-012). Fieldwork was permitted by the Prince Edward Islands Management committee.

Data accessibility. Data and R scripts supporting this manuscript can be found at https://github.com/ryanreisinger/giantPetrels (doi:10.5281/zenodo.3984174) [87] and at Dryad Digital Repository (doi:10.5061/dryad.31zcrjdj0) [88].

Authors' contributions. Conceptualization: R.R.R. and P.A.P.; fieldwork: T.C.-K.; Laboratory work: T.C.-K. and M.C.; analyses: R.R.R., T.C.-K., M.C. and Y.C.; writing—original draft preparation: R.R.R., T.C.-K. and M.C.; writing—review and editing: R.R.R., T.C.-K., M.C., Y.C. and P.A.P.

Competing interests. We have no competing interests.

Funding. This work was funded by the South African National Research Foundation, under the South African National Antarctic Program (grant no. SNA93071 to P.A.P.) and the South African Network for Coastal and Oceanic Research (grant no. 94916 to R.R.R.).

Acknowledgements. We thank the South African Department of Environmental Affairs for logistical support. We are grateful to John Lanham and Ian Newton (University of Cape Town) for the stable isotope analyses and to Jess Berndt and David Green for fieldwork. We appreciate the comments of several reviewers, which improved this paper.

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
