## [Reviewer comments · Royal Society Open Science]

Review History

RSOS-200649.R0 (Original submission)

Review form: Reviewer 1

Is the manuscript scientifically sound in its present form?

Yes

Are the interpretations and conclusions justified by the results?

Yes

Is the language acceptable?

Yes

Do you have any ethical concerns with this paper?

Yes

Have you any concerns about statistical analyses in this paper?

Yes

Recommendation?

Major revision is needed (please make suggestions in comments)

Comments to the Author(s)

Overall this paper is well-written and coherent. Conclusions followed well from the evidence in the text and were appropriately drawn in the context segregation of foraging and habitat use segregation as a consequence of competition avoidance. Support from the literature was sufficient to bolster claims and references were appropriately selected. However I have some methodological question described below.

See comments below:

L31: You use characterise here and along the text characterize. Please, try to be coherent on this.

L49-57: Why do you do not show also studies on seabirds to demonstrate the segregation both in space and time for sympatric species? It would be good to see some examples on seabirds.

L127: The default conclusion of non-overlap in seabird studies is most often intra-specific competition, thereby perpetuating an assumption that has little or no support in my opinion. One must demonstrate resource limitation and competitive exclusion to confirm competition, which is rarely possible with seabird studies (especially from remote tracking). There are other equally plausible explanations to spatial and habitat segregation in seabirds that you could explore here.

L134-135: Could you explain more in detail the oceanography of the study area? Such intermediate position in relation to frontal zones should have particular oceanographic features that may or not limit or explain the activity and distribution patterns both species and sexes.

L138-140: The last censuses for these populations were in 2009? It would be good to see recent numbers. Plus, regarding populations numbers did you thought in relate the foraging patterns with the population size to analyse to what extent foraging effort change according to the population size?

L178: Please explain the unit of h factor. Degrees or meters? Moreover, why you used an h factor of 1.0? You could have followed the methodology in Lascelles BG, Taylor PR, Miller MGR, Dias MP, Opper S, Torres L, Hedd A, Le Corre M, Phillips RA, Shaffer SA, Weimerskirch H, Small C (2016) Applying global criteria to tracking data to define important areas for marine conservation. *Divers Distrib* 22:422–431. <https://doi.org/10.1111/ddi.12411>. In this way, the mean ARS behaviour for all the foraging trips in each year could be used separately as h factor and you could also use the average ARS for each individual per year as a response variable. Perhaps females show higher ARS scale as a response to the longer and widespread oceanic trips?

L177-179: Moreover, one thing that I noticed is that you used all the positions to generate utilisation distributions and do not filtered the positions where birds engage in ARS or searching behaviour. Do you considered filtering the positions based on speed thresholds or to use clustering analysis (e.g. EMBC - Expectation-Maximization Binary Clustering) or Hidden Markov Models to select the positions indicative of searching behaviour and measure the kernel overlap over these positions only? - I really think that this study would benefit from investigating the habitat use by measuring the spatial overlap only in the foraging locations where birds initiate searching behaviour.

Review form: Reviewer 2

Is the manuscript scientifically sound in its present form?

Yes

Are the interpretations and conclusions justified by the results?

Yes

Is the language acceptable?

No

Do you have any ethical concerns with this paper?

No

Have you any concerns about statistical analyses in this paper?

No

Recommendation?

Major revision is needed (please make suggestions in comments)

Comments to the Author(s)

This manuscript provides a good description of how northern and southern giant petrels partition trophic resources and spatially at Marion Island. The authors present three years of tracking data and associated stable isotope values from plasma (short-term concurrent time period). They find that females of the two species have distinct distributions at sea, while the males forage closer to shore or on land. All four categories had low overlap in their isotopic niches, but when just considering birds that made at-sea distant trips male and female NGPs isotopic niches overlap. I have numerous questions about interpretation, structure, and clarity, but I found the study informative.

Main comments

An indication of how “good” your study years would be helpful in informing the discussion. Potentially this could be a reproductive success, breeding number, or a metric of carcass numbers. While not perfect this could help provide context for how abundant prey was, the degree of competition, and how pressing the need might be to exploit other resources – this seems most relevant for the males.

Annual differences or similarities are not addressed. As annual variation is often a strong driver of seabird diets and distributions please provide some justification for why you decided to pool the years.

In the introduction the following hypothesis is given “Given the species’ recent divergence and marked sexual dimorphism, we predict that segregation between sexes should be greater than that between species, but that the two species should also show some segregation along one or more niche-axes, which has developed in the 0.2 million years since their divergence.” This seems to generally hold true, but the very distinct geographic distributions of the females of the two species seem more distinct than the male-female segregation particularly for the NGPs. Returning to this in the discussion would be helpful.

While great for completeness, this manuscript relies heavily on supplemental figures and tables and at times that is distracting for the reader. The stable isotope analysis section has three tables and two figures devoted to it in the main text (not supplemental). Likewise the two figures with maps in the main text seem somewhat redundant. I suggest balancing the data shown in the main manuscript and reducing the reliance on supplemental materials.

In line Comments:

L50 prey type, size, or?

L51 remove internal parenthesis – issue throughout the paper

L58 “structures”

L45-59 The examples in this paragraph are interesting, but they feel like a list of disparate examples of the many permutations of differences in diet/space/time between similar species. A better description of the linking concepts is needed, particularly to understand why the authors chose to highlight these examples. The first sentence and the concluding sentence of the paragraph seem to be expressing the same concept.

L61-68 Are we choosing between the ‘forage selection hypothesis’ and the ‘activity budget hypothesis’? “Among the explanations” implies that there are other explanations not discussed here. This paragraph needs editing to clearly highlight how males and females may or may not partition resources.

L70-86 These seem like one paragraph rather than two.

L112: given the large body of work on this topic, “for the first time” seems like an overstatement and is not a strong or informative justification of this study.

L119: reference needed for “0.2 million years...”

L123-24: It would be useful to introduce stable isotopes earlier in the introduction.

L126: Given the number of tracking/stable isotope studies of just seabirds, “novel” seems out of place here.

L138: Since your study site is on Marion Island it would be useful to know how many of these breeding pairs are on that island.

L143: This implies that if they had a nesting location and phenology similar to NGP, but a bill tip of SGP they would be classed as NGP?

L148: Handling times would be a useful thing to report.

L156: Why/how did you choose the other 22 birds for molecular sexing?

L170: Will you archive this and create a doi for the repository? Thank you for including your codes!

L168: The sampling interval of the GPS tags should be reported at the beginning of this section.

L172: Did this filtering remove locations?

L178: Units for h?

L182-185: This method has been used before. It would be good to include a citation or two.

L187: You need to include general details of the seven environmental data chosen. You could simply say ‘sea surface temperature’ and then use the supplementary table to provide the source of the data. Move the list of variables in L196-197 to the top of the paragraph.

L187: I guess these variables are not for the locations on land and just for the at-sea distributions? Specify “at-sea habitat.” Potentially incorporating land variables would be a useful approach for the male giant petrels.

L206-208: It is unclear which values you report.

L225: Does this mean your sample size was 90 or less than 90?

L227: Moving (Mardia 1970) to just before the comma would make this sentence easier to read.

L236: Since some individuals clearly foraged locally (at a known latitude), how does this correction factor influence your results? This needs to be explained and justified. You could be correcting for a combined penguin, seal, burrowing petrel signal?

L246: It seems a bit backwards to report your guess for the sexes and then confirm them with molecular sexing. Why not say first that your molecular sexing confirmed your grouping and then report?

L252: You should provide some indication of how long birds were tracked for. 1 day, 5 days, 2 weeks??

L253: You don't provide any quantitative methods for how you determined these two foraging strategies. These need to be included. Why do the females fit into this group if they are the outliers? It seems like ~90km is too far to be foraging on penguin/seal resources or in-shore waters around Marion?

L253: Were birds that were categorized in the second group tracked for longer? I.e. Long enough to make both the short trips observed in the first group AND a long trip?

L281: Figure 3 is hard to interpret since the points are plotted on top each other.

L288: It isn't surprising that at the 50% Utilization distribution scale that the males wouldn't be segregated. Without knowing the unit of the smoothing factor h it is hard to know how "smoothed" these distributions are relative to the data, but from your point data it seems like they might be very smoothed. Did you see finer scale clustering and separation between the males in the GPS data or was your GPS data too low temporal resolution to tell?

L328: I am not sure that you explicitly demonstrate this with your data.

L329/L332 The statements "exploiting the same resources" and "males were isotopically segregated" seem contradictory.

L338 replace "utilized" with "used"

L348-351: You included oceanographic habitat analysis in this paper. How do your results support this statement?

L354 Integrating the observations of feeding with the first paragraph would be useful.

L359 Is this because the SGP go further south or are just foraging on other species that range further south? Since you are looking at plasma it doesn't seem like the first would fit with your tracking data results. Clarify and integrate the tracking into this discussion.

L364: I think your methods may only resolve "large scale spatial overlap", i.e. you show that they are both foraging on the same island. In Figure 1 the NGPs that made short trips seem to have a much more extensive at sea distribution than the SGPs. They also seem to visit different beaches that the SGPs (that were tracked) didn't visit.

L365: More details (and analysis?) are needed to explain how the timing of spatial overlap is important in order support these statements.

L373-383 Since you can't directly address this, shortening this explanation and integrating it into the prior paragraph would be helpful. Do you know if there were ample carcasses available during your study years?

L385 You Supplemental Figure 7 shows that you are missing the prey with the elevated N. Without options (and data) for your high N prey items your prey paragraph feels very speculative. I am skeptical about burrowing petrels being a high component of the diet of male giant petrels – do you have any observations from your study years?

L405-407 The Gonzalis-Solis and Granroth-Wilding study were both from Bird Island, South Georgia so it seems odd to contrast them to each other like they were from different places. It seems likely that annual variation plays a role as well. Did your study find any variation between the three years?

L410: Both males and females forage at sea during the winter, right? Perhaps males exclude females from easily accessed carrion resources, but are able to supplement with at-sea resources as needed.

L419 The reference at the end of this sentence seems out of place. Perhaps a review would make more sense.

L417-432 This paragraph is unexpected and out of place given how the study was framed and the hypothesis presented in the introduction. Omit or reframe to include anthropogenic threats in the introduction.

L435 While it may be true “for the first time” doesn't really speak to the interesting or informative parts of this study. Omit.

Review form: Reviewer 3 (Andrea Raya Rey)

Is the manuscript scientifically sound in its present form?

Yes

Are the interpretations and conclusions justified by the results?

Yes

Is the language acceptable?

Yes

Do you have any ethical concerns with this paper?

No

Have you any concerns about statistical analyses in this paper?

No

Recommendation?

Accept with minor revision (please list in comments)

Comments to the Author(s)

studying intra and interspecific segregation in seabirds is key to understand theoretical aspects of coexistence in sympatric sibling species but also in terms of species conservation.

This study investigated segregation in two sympatric sibling species using tracking, environmental and stable isotope data. The study is well executed and well written.

I suggest only minor changes to clarify certain aspects.

1. in the conclusion the authors suggest very nicely how this intra and inter specific segregation arose in evolutionary times. however, at the beginning and within the introduction they focused on the common definition of competition to address segregation, which implies limited resources which is not prove neither imply in this study. I suggest adding other theories for segregation within the introduction that may be influencing in the pattern found in the study spp (i.e. life history theory).
2. I found very interesting exploring habitat use using random forest analysis, I suggest audience may be benefit if further details are given for the analyses. Also, more details in the results section regarding this will be useful.
3. Lines 349-351 what do the authors mean this this statement? How is the interpretation of oceanography influencing the foraging strategy? diet preference could influence foraging strategy and thus oceanography as prey rely on certain aspects of the ocean, please clarify.

Decision letter (RSOS-200649.R0)

Dear Dr Reisinger,

The editors assigned to your paper ("Foraging behaviour and habitat-use drives niche segregation in sibling seabird species") have now received comments from reviewers. We would like you to revise your paper in accordance with the referee and Associate Editor suggestions which can be found below (not including confidential reports to the Editor). Please note this decision does not guarantee eventual acceptance.

Please submit a copy of your revised paper before 17-Jul-2020. Please note that the revision deadline will expire at 00.00am on this date. If we do not hear from you within this time then it will be assumed that the paper has been withdrawn. In exceptional circumstances, extensions may be possible if agreed with the Editorial Office in advance. We do not allow multiple rounds of revision so we urge you to make every effort to fully address all of the comments at this stage. If deemed necessary by the Editors, your manuscript will be sent back to one or more of the original reviewers for assessment. If the original reviewers are not available, we may invite new reviewers.

- Data accessibility

<http://datadryad.org/submit?journalID=RSOS&manu=RSOS-200649>

- Competing interests

- Authors' contributions

- Acknowledgements

- Funding statement

on behalf of Prof Pete Smith (Subject Editor)
openscience@royalsociety.org

Associate Editor's comments:

Given the commentary supplied by the reviewers, the Editors would like you to fully address the referees' concerns. Bear in mind that the journal does not generally permit multiple rounds of revision, and if you are not able to satisfy the critical reviewers that the paper is ready for publication on receipt of the revision, we may not be able to proceed further, so please do your best to engage critically with the comments and remember to supply a point-by-point response as well as a 'tracked changes' version of the revised paper. We'll look forward to receiving your revision in due course.

Reviewers' Comments to Author:

Reviewer: 1

Comments to the Author(s)

Overall this paper is well-written and coherent. Conclusions followed well from the evidence in the text and were appropriately drawn in the context segregation of foraging and habitat use segregation as a consequence of competition avoidance. Support from the literature was sufficient to bolster claims and references were appropriately selected. However I have some methodological question described below.

See comments below:

L31: You use characterise here and along the text characterize. Please, try to be coherent on this.

L49-57: Why do you do not show also studies on seabirds to demonstrate the segregation both in space and time for sympatric species? It would be good to see some examples on seabirds.

L127: The default conclusion of non-overlap in seabird studies is most often intra-specific competition, thereby perpetuating an assumption that has little or no support in my opinion. One must demonstrate resource limitation and competitive exclusion to confirm competition, which is rarely possible with seabird studies (especially from remote tracking). There are other equally plausible explanations to spatial and habitat segregation in seabirds that you could explore here.

L134-135: Could you explain more in detail the oceanography of the study area? Such intermediate position in relation to frontal zones should have particular oceanographic features that may or not limit or explain the activity and distribution patterns both species and sexes.

L138-140: The last censuses for these populations were in 2009? It would be good to see recent numbers. Plus, regarding populations numbers did you thought in relate the foraging patterns with the population size to analyse to what extent foraging effort change according to the population size?

L178: Please explain the unit of h factor. Degrees or meters? Moreover, why you used an h factor of 1.0? You could have followed the methodology in Lascelles BG, Taylor PR, Miller MGR, Dias MP, Oppel S, Torres L, Hedd A, Le Corre M, Phillips RA, Shaffer SA, Weimerskirch H, Small C (2016) Applying global criteria to tracking data to define important areas for marine conservation. *Divers Distrib* 22:422–431. <https://doi.org/10.1111/ddi.12411>. In this way, the mean ARS behaviour for all the foraging trips in each year could be used separately as h factor and you

could also use the average ARS for each individual per year as a response variable. Perhaps females show higher ARS scale as a response to the longer and widespread oceanic trips?

L177-179: Moreover, one thing that I noticed is that you used all the positions to generate utilisation distributions and do not filtered the positions where birds engage in ARS or searching behaviour. Do you considered filtering the positions based on speed thresholds or to use clustering analysis (e.g. EMBC - Expectation-Maximization Binary Clustering) or Hidden Markov Models to select the positions indicative of searching behaviour and measure the kernel overlap over these positions only? - I really think that this study would benefit from investigating the habitat use by measuring the spatial overlap only in the foraging locations where birds initiate searching behaviour.

Reviewer: 2

Comments to the Author(s)

This manuscript provides a good description of how northern and southern giant petrels partition trophic resources and spatially at Marion Island. The authors present three years of tracking data and associated stable isotope values from plasma (short-term concurrent time period). They find that females of the two species have distinct distributions at sea, while the males forage closer to shore or on land. All four categories had low overlap in their isotopic niches, but when just considering birds that made at-sea distant trips male and female NGPs isotopic niches overlap. I have numerous questions about interpretation, structure, and clarity, but I found the study informative.

Main comments

An indication of how “good” your study years would be helpful in informing the discussion. Potentially this could be a reproductive success, breeding number, or a metric of carcass numbers. While not perfect this could help provide context for how abundant prey was, the degree of competition, and how pressing the need might be to exploit other resources – this seems most relevant for the males.

Annual differences or similarities are not addressed. As annual variation is often a strong driver of seabird diets and distributions please provide some justification for why you decided to pool the years.

In the introduction the following hypothesis is given “Given the species’ recent divergence and marked sexual dimorphism, we predict that segregation between sexes should be greater than that between species, but that the two species should also show some segregation along one or more niche-axes, which has developed in the 0.2 million years since their divergence.” This seems to generally hold true, but the very distinct geographic distributions of the females of the two species seem more distinct than the male-female segregation particularly for the NGPs. Returning to this in the discussion would be helpful.

While great for completeness, this manuscript relies heavily on supplemental figures and tables and at times that is distracting for the reader. The stable isotope analysis section has three tables and two figures devoted to it in the main text (not supplemental). Likewise the two figures with maps in the main text seem somewhat redundant. I suggest balancing the data shown in the main manuscript and reducing the reliance on supplemental materials.

In line Comments:

L50 prey type, size, or?

L51 remove internal parenthesis – issue throughout the paper

L58 “structures”

L45-59 The examples in this paragraph are interesting, but they feel like a list of disparate examples of the many permutations of differences in diet/space/time between similar species. A better description of the linking concepts is needed, particularly to understand why the authors chose to highlight these examples. The first sentence and the concluding sentence of the paragraph seem to be expressing the same concept.

L61-68 Are we choosing between the ‘forage selection hypothesis’ and the ‘activity budget hypothesis’? “Among the explanations” implies that there are other explanations not discussed here. This paragraph needs editing to clearly highlight how males and females may or may not partition resources.

L70-86 These seem like one paragraph rather than two.

L112: given the large body of work on this topic, “for the first time” seems like an overstatement and is not a strong or informative justification of this study.

L119: reference needed for “0.2 million years...”

L123-24: It would be useful to introduce stable isotopes earlier in the introduction.

L126: Given the number of tracking/stable isotope studies of just seabirds, “novel” seems out of place here.

L138: Since your study site is on Marion Island it would be useful to know how many of these breeding pairs are on that island.

L143: This implies that if they had a nesting location and phenology similar to NGP, but a bill tip of SGP they would be classed as NGP?

L148: Handling times would be a useful thing to report.

L156: Why/how did you choose the other 22 birds for molecular sexing?

L170: Will you archive this and create a doi for the repository? Thank you for including your codes!

L168: The sampling interval of the GPS tags should be reported at the beginning of this section.

L172: Did this filtering remove locations?

L178: Units for h?

L182-185: This method has been used before. It would be good to include a citation or two.

L187: You need to include general details of the seven environmental data chosen. You could simply say ‘sea surface temperature’ and then use the supplementary table to provide the source of the data. Move the list of variables in L196-197 to the top of the paragraph.

L187: I guess these variables are not for the locations on land and just for the at-sea distributions? Specify “at-sea habitat.” Potentially incorporating land variables would be a useful approach for the male giant petrels.

L206-208: It is unclear which values you report.

L225: Does this mean your sample size was 90 or less than 90?

L227: Moving (Mardia 1970) to just before the comma would make this sentence easier to read.

L236: Since some individuals clearly foraged locally (at a known latitude), how does this correction factor influence your results? This needs to be explained and justified. You could be correcting for a combined penguin, seal, burrowing petrel signal?

L246: It seems a bit backwards to report your guess for the sexes and then confirm them with molecular sexing. Why not say first that your molecular sexing confirmed your grouping and then report?

L252: You should provide some indication of how long birds were tracked for. 1 day, 5 days, 2 weeks??

L253: You don't provide any quantitative methods for how you determined these two foraging strategies. These need to be included. Why do the females fit into this group if they are the outliers? It seems like ~90km is too far to be foraging on penguin/seal resources or in-shore waters around Marion?

L253: Were birds that were categorized in the second group tracked for longer? I.e. Long enough to make both the short trips observed in the first group AND a long trip?

L281: Figure 3 is hard to interpret since the points are plotted on top each other.

L288: It isn't surprising that at the 50% Utilization distribution scale that the males wouldn't be segregated. Without knowing the unit of the smoothing factor h it is hard to know how "smoothed" these distributions are relative to the data, but from your point data it seems like they might be very smoothed. Did you see finer scale clustering and separation between the males in the GPS data or was your GPS data too low temporal resolution to tell?

L328: I am not sure that you explicitly demonstrate this with your data.

L329/L332 The statements "exploiting the same resources" and "males were isotopically segregated" seem contradictory.

L338 replace "utilized" with "used"

L348-351: You included oceanographic habitat analysis in this paper. How do your results support this statement?

L354 Integrating the observations of feeding with the first paragraph would be useful.

L359 Is this because the SGP go further south or are just foraging on other species that range further south? Since you are looking at plasma it doesn't seem like the first would fit with your tracking data results. Clarify and integrate the tracking into this discussion.

L364: I think your methods may only resolve "large scale spatial overlap", i.e. you show that they are both foraging on the same island. In Figure 1 the NGPs that made short trips seem to have a much more extensive at sea distribution than the SGPs. They also seem to visit different beaches that the SGPs (that were tracked) didn't visit.

L365: More details (and analysis?) are needed to explain how the timing of spatial overlap is important in order to support these statements.

L373-383 Since you can't directly address this, shortening this explanation and integrating it into the prior paragraph would be helpful. Do you know if there were ample carcasses available during your study years?

L385 You Supplemental Figure 7 shows that you are missing the prey with the elevated N. Without options (and data) for your high N prey items your prey paragraph feels very speculative. I am skeptical about burrowing petrels being a high component of the diet of male giant petrels – do you have any observations from your study years?

L405-407 The Gonzalis-Solis and Granroth-Wilding study were both from Bird Island, South Georgia so it seems odd to contrast them to each other like they were from different places. It seems likely that annual variation plays a role as well. Did your study find any variation between the three years?

L410: Both males and females forage at sea during the winter, right? Perhaps males exclude females from easily accessed carrion resources, but are able to supplement with at-sea resources as needed.

L419 The reference at the end of this sentence seems out of place. Perhaps a review would make more sense.

L417-432 This paragraph is unexpected and out of place given how the study was framed and the hypothesis presented in the introduction. Omit or reframe to include anthropogenic threats in the introduction.

L435 While it may be true “for the first time” doesn't really speak to the interesting or informative parts of this study. Omit.

Reviewer: 3

Comments to the Author(s)

studying intra and interspecific segregation in seabirds is key to understand theoretical aspects of coexistence in sympatric sibling species but also in terms of species conservation. This study investigated segregation in two sympatric sibling species using tracking, environmental and stable isotope data. The study is well executed and well written.

I suggest only minor changes to clarify certain aspects.

1. in the conclusion the authors suggest very nicely how this intra and inter specific segregation arose in evolutionary times. however, at the beginning and within the introduction they focused on the common definition of competition to address segregation, which implies limited resources which is not prove neither imply in this study. I suggest adding other theories for segregation within the introduction that may be influencing in the pattern found in the study spp (i.e. life history theory).
2. I found very interesting exploring habitat use using random forest analysis, I suggest audience may be benefit if further details are given for the analyses. Also, more details in the results section regarding this will be useful.
3. Lines 349-351 what do the authors mean this this statement? How is the interpretation of oceanography influencing the foraging strategy? diet preference could influence foraging strategy and thus oceanography as prey rely on certain aspects of the ocean, please clarify.

Author's Response to Decision Letter for (RSOS-200649.R0)

See Appendix A.

RSOS-200649.R1 (Revision)

Review form: Reviewer 2

Is the manuscript scientifically sound in its present form?

Yes

Are the interpretations and conclusions justified by the results?

Yes

Is the language acceptable?

Yes

Do you have any ethical concerns with this paper?

Yes

Have you any concerns about statistical analyses in this paper?

No

Recommendation?

Accept with minor revision (please list in comments)

Comments to the Author(s)

I found the manuscript much improved. Thank you for your detailed replies to the reviewer comments.

The authors have addressed the potential for annual differences in the permutation analysis, but it is unclear how annual differences were accounted for (or not) in the at-sea habitat use analysis or the stable isotope statistical analysis. If years were not accounted for this should be clearly stated in the text. Otherwise, my comments, listed below are very minor.

L62 insert "same" three axes...

L68 consider commas rather than parenthesis

L123 "high resolution tracks" at hourly locations this "high-resolution" statement seems like an oversight. Omit

Replace utilize with use throughout the manuscript

L126: Rephrase: The conclusion of your introduction reads like a result "we show segregation in their habitat use and foraging ecology that may have evolved as a competition avoidance mechanism. Consider omitting this last statement

L132 Rephrase (missing comma?) "are located on a rise in water ~5000 m deep"

L132 Add the word "habitat" before variability

L154: Handling time should be reported as a value \pm standard deviation.

L184: This sentence may not be needed: "We tested other class classification methods, and bivariate k-means, but these results were not ecologically sensible."

L260: Rephrase for grammar: "as far as possible"

L320-322: Rewrite to avoid 1 sentence paragraphs.

L321: Reverse order here for consistency: NGPs and SGPs

L336: Reiterate that this is during incubation.

Review form: Reviewer 3 (Andrea Raya Rey)

Is the manuscript scientifically sound in its present form?

Yes

Are the interpretations and conclusions justified by the results?

Yes

Is the language acceptable?

Yes

Do you have any ethical concerns with this paper?

No

Have you any concerns about statistical analyses in this paper?

No

Recommendation?

Accept as is

Comments to the Author(s)

Authors has revised and added all the suggestions in the first round, I believe the manuscript is sound and sum to the knowledge on sibling species competition and segregation

Decision letter (RSOS-200649.R1)

Dear Dr Reisinger,

On behalf of the Editors, we are pleased to inform you that your Manuscript RSOS-200649.R1 "Foraging behaviour and habitat-use drives niche segregation in sibling seabird species" has been accepted for publication in Royal Society Open Science subject to minor revision in accordance with the referees' reports. Please find the referees' comments along with any feedback from the Editors below my signature.

Please submit your revised manuscript and required files (see below) no later than 7 days from today's (ie 12-Aug-2020) date. Note: the ScholarOne system will 'lock' if submission of the revision is attempted 7 or more days after the deadline. If you do not think you will be able to meet this deadline please contact the editorial office immediately.

Please note article processing charges apply to papers accepted for publication in Royal Society Open Science (<https://royalsocietypublishing.org/rsos/charges>). Charges will also apply to papers transferred to the journal from other Royal Society Publishing journals, as well as papers submitted as part of our collaboration with the Royal Society of Chemistry

(<https://royalsocietypublishing.org/rsos/chemistry>). Fee waivers are available but must be requested when you submit your revision (<https://royalsocietypublishing.org/rsos/waivers>).

Best regards,

on behalf of the Associate Editor and Professor Pete Smith (Subject Editor)
openscience@royalsociety.org

Associate Editor Comments to Author:

A few minor changes remain to be made, but the reviewers are largely satisfied your manuscript is ready for publication - congratulations. We look forward to receiving your final version soon.

Reviewer comments to Author:

Reviewer: 2

Comments to the Author(s)

I found the manuscript much improved. Thank you for your detailed replies to the reviewer comments.

The authors have addressed the potential for annual differences in the permutation analysis, but it is unclear how annual differences were accounted for (or not) in the at-sea habitat use analysis or the stable isotope statistical analysis. If years were not accounted for this should be clearly stated in the text. Otherwise, my comments, listed below are very minor.

L62 insert "same" three axes...

L68 consider commas rather than parenthesis

L123 "high resolution tracks" at hourly locations this "high-resolution" statement seems like an oversight. Omit

Replace utilize with use throughout the manuscript

L126: Rephrase: The conclusion of your introduction reads like a result "we show segregation in their habitat use and foraging ecology that may have evolved as a competition avoidance mechanism. Consider omitting this last statement

L132 Rephrase (missing comma?) "are located on a rise in water ~5000 m deep"

L132 Add the word "habitat" before variability

L154: Handling time should be reported as a value \pm standard deviation.

L184: This sentence may not be needed: "We tested other class classification methods, and bivariate k-means, but these results were not ecologically sensible."

L260: Rephrase for grammar: "as far as possible"

L320-322: Rewrite to avoid 1 sentence paragraphs.

L321: Reverse order here for consistency: NGPs and SGPs

L336: Reiterate that this is during incubation.

Reviewer: 3

Comments to the Author(s)

Authors has revised and added all the suggestions in the first round, I believe the manuscript is sound and sum to the knowledge on sibling species competition and segregation

===PREPARING YOUR MANUSCRIPT===

- one version identifying all the changes that have been made (for instance, in coloured highlight, in bold text, or tracked changes);
- a 'clean' version of the new manuscript that incorporates the changes made, but does not highlight them.

This version will be used for typesetting.

===PREPARING YOUR REVISION IN SCHOLARONE===

<https://royalsociety.org/journals/authors/author-guidelines/#supplementary-material> to include a suitable title and informative caption. An example of appropriate titling and captioning may be found at https://figshare.com/articles/Table_S2_from_Is_there_a_trade-off_between_peak_performance_and_performance_breadth_across_temperatures_for_aerobic_scops_in_teleost_fishes_/3843624.

Author's Response to Decision Letter for (RSOS-200649.R1)

See Appendix B.

Decision letter (RSOS-200649.R2)

Dear Dr Reisinger,

It is a pleasure to accept your manuscript entitled "Foraging behaviour and habitat-use drives niche segregation in sibling seabird species" in its current form for publication in Royal Society Open Science. The comments of the reviewer(s) who reviewed your manuscript are included at the foot of this letter.

Please ensure that you send to the editorial office an editable version of your accepted manuscript, and individual files for each figure and table included in your manuscript. You can

send these in a zip folder if more convenient. Failure to provide these files may delay the processing of your proof. You may disregard this request if you have already provided these files to the editorial office.

on behalf of Prof Pete Smith (Subject Editor)
openscience@royalsociety.org

Appendix A

Associate Editor's comments:

Given the commentary supplied by the reviewers, the Editors would like you to fully address the referees' concerns. Bear in mind that the journal does not generally permit multiple rounds of revision, and if you are not able to satisfy the critical reviewers that the paper is ready for publication on receipt of the revision, we may not be able to proceed further, so please do your best to engage critically with the comments and remember to supply a point-by-point response as well as a 'tracked changes' version of the revised paper. We'll look forward to receiving your revision in due course.

RESPONSE: We thank the Associate Editor and Reviewers for their thoughtful remarks. We address each of the reviewer's comments in detail below, but broadly we have made the following major changes:

We focus less on current competition as the force driving segregation, acknowledging that other mechanisms, mentioned in the introduction now, can result in the observed segregation. However, we note that a different causal mechanism does not negate the patterns that can be observed in our data.

We have updated the methods substantially in response to the reviewers' suggestions. At the same time, we provide more details on the methodology, as requested. Particularly our revisions focussed on the recalculation of the utilisation distributions. We note that this has not qualitatively changed our results.

Finally, we allow for the effect of potential inter-annual effects by constraining our permutation tests of spatial and isotopic overlap by year. A full analysis relating any year-to-year variations in segregation that there may be (which are not obvious to us from exploratory analysis or the current analysis) is beyond the scope of the work, since the required data (year-to-year population size information and prey abundance information) is not available to us, and in our opinion would be difficult to collect properly. However, we briefly discuss this question in the Discussion and recommend such work in the Conclusion.

As well as the revised document, we have attached a marked-up document showing the changes (compared with references unformatted, for simplicity). Line numbers below refer to the clean, reference-formatted text, but we quote the full references for readability.

Reviewers' Comments to Author:

Reviewer: 1

Overall this paper is well-written and coherent. Conclusions followed well from the evidence in the text and were appropriately drawn in the context segregation of foraging and habitat use segregation as a consequence of competition avoidance. Support from the literature was sufficient to bolster claims and references were appropriately selected. However I have some methodological question described below.

See comments below:

L31: You use characterise here and along the text characterize. Please, try to be coherent on this.

RESPONSE: We have made sure to use British spelling throughout.

L49-57: Why do you do not show also studies on seabirds to demonstrate the segregation both in space and time for sympatric species? It would be good to see some examples on seabirds.

RESPONSE: We note that Reviewer 2 found the diverse examples in the introduction interesting. To maintain the broader interest of the paper and not be too taxonomically restrictive, we have changed only one of the examples (allochrony, or segregation along a time axis) for a seabird example. However, we now state at the beginning of paragraph three, that inter- and intra-specific segregation is frequently described in seabirds, with example references (see response to Reviewer 2 below), as a link to the brief introduction to the tools we used in this study.

L127: The default conclusion of non-overlap in seabird studies is most often intra-specific competition, thereby perpetuating an assumption that has little or no support in my opinion. One must demonstrate resource limitation and competitive exclusion to confirm competition, which is rarely possible with seabird studies (especially from remote tracking). There are other equally plausible explanations to spatial and habitat segregation in seabirds that you could explore here.

RESPONSE: We agree that it is often not possible to demonstrate resource limitation and competitive exclusion, as is the case in our study, and there may be alternative explanations for segregation. We have thus included this caveat in our introduction:

“Competition (whether current or in species’ evolutionary history) is frequently invoked as an explanation for niche segregation, but segregation may result from other factors including non-competitive evolutionary adaptation and stochastic processes (Begon et al. 2006).” [L66-68]

Further, our revised statement introduction about the hypothesis for sexual segregation makes it clearer that segregation can be due to intrinsic constraints rather than competition avoidance:

“The ‘specialisation hypothesis’ suggests that males and females segregate habitats and/or diets due to different requirements, opportunities and constraints related to their morphology, physiology and reproductive roles (Catry & Phillips 2005, Wearmouth & Sims 2008)”. [L64-66]

L134-135: Could you explain more in detail the oceanography of the study area? Such intermediate position in relation to frontal zones should have particular oceanographic features that may or not limit or explain the activity and distribution patterns both species and sexes.

RESPONSE: We have added additional information on the local oceanography:

“The Prince Edward Islands (46.9° S, 37.7° E), comprising Marion Island and Prince Edward Island, are located on a rise in water ~5000 m deep. There is high mesoscale variability in the vicinity of the islands, stemming from the presence of eddies spawned as the west-flowing Antarctic Circumpolar Current interacts with the South-

West Indian Ridge upstream of the islands. The islands are situated in the Polar Frontal Zone, which is delineated by the Antarctic Polar Front in the south and the Subantarctic Front in the north (Figure 1) (Ansorge & Lutjeharms 2002) – areas attractive to various marine predators (Reisinger, Raymond, et al. 2018). The biophysical characteristics of the surrounding ocean thus show strong latitudinal patterns”. [L129-135]

L138-140: The last censuses for these populations were in 2009? It would be good to see recent numbers. Plus, regarding populations numbers did you thought in relate the foraging patterns with the population size to analyse to what extent foraging effort change according to the population size?

RESPONSE: We have added the most recent information that is available to us, as well as information on trends.

“In 2013/2014, Marion Island had about 443 breeding pairs of NGPs (increasing since the early 2000s) and about 1,583 breeding pairs of SGPs (stable since the late 1990s) (Ryan et al. 2009, ACAP 2020). Together with Prince Edward Island, this represents around 6% of their respective global populations (Ryan et al. 2009).” [L137-140]

Regarding an analysis relating population size to foraging effort, this would require estimating maps of habitat use and then converting these to density maps or spatially-explicit abundance estimate for each species. This is not necessarily a straightforward conversion according to population size (which data was not available to us for the study years anyway). Given this, and that the overlap and segregation patterns are consistent across years (see response to one of Reviewer 2’s main comments), we believe such an analysis would not be informative with respect to the stated goals of this manuscript, although it would certainly be interesting to explore with additional data in a future, more detailed manuscript. However, we have added a very brief discussion of this issue (see response to Reviewers 2).

L178: Please explain the unit of h factor. Degrees or meters? Moreover, why you used an h factor of 1.0? You could have followed the methodology in Lascelles BG, Taylor PR, Miller MGR, Dias MP, Opper S, Torres L, Hedd A, Le Corre M, Phillips RA, Shaffer SA, Weimerskirch H, Small C (2016) Applying global criteria to tracking data to define important areas for marine conservation. *Divers Distrib* 22:422–431. <https://doi.org/10.1111/ddi.12411>. In this way, the mean ARS behaviour for all the foraging trips in each year could be used separately as h factor and you could also use the average ARS for each individual per year as a response variable. Perhaps females show higher ARS scale as a response to the longer and widespread oceanic trips?

RESPONSE: The h value has units of degrees; we have added this information. As we stated in the text, we attempted to use the least squares cross-validation method to select h values, but the process failed to converge as is often the case (see documentation for *kernelUD* function in the *adehabitatHR* package). We therefore used the ‘ad-hoc’ method of choosing an initially high h value (oversmoothing, positive bias) and then incrementally decreasing it until the utilisation distribution

starts to break up (undersmoothing, negative bias) (Gitzen et al. 2006, DOI: 10.2193/0022-541X(2006)70[1334:BSFFAO]2.0.CO;2 ; Schuler et al. 2014, DOI: 10.2981/wlb.12117). This method has been shown to perform better than the 'h-ref' method (Schuler et al. 2014, DOI: 10.2981/wlb.12117), which we also found to oversmooth the distribution – this is particularly important in the context of Reviewer 2's comments about the smooth distributions. To try and avoid issues related to choosing h, during our revisions we calculated utilisation distributions using the 'akde' method (Fleming et al. 2015, DOI: 10.1890/14-2010.1), which takes into account the autocorrelation inherent to tracking data and does not require an h value to be specified. However, we found this method to oversmooth the distributions unacceptably, as noted by Péron (2019, DOI: 10.1186/s40462-019-0161-9).

Regarding the method used by Lascelles et al. 2016, we did not want the uncertainties associated with behavioural classification along the track (see response to next comment) to unduly influence the h value estimation. Thus we still use values chosen using the 'ad-hoc' method, but we explain this more clearly in the text, have used a separate value for males making short trips (h = 0.005 degrees), and chose a slightly smaller value for animals making long trips (h = 0.8 versus 1.0 degrees).

Regarding using ARS values as a response variable, our current random forest classification uses the group (sex by species combinations) as the response variable, to investigate how environmental conditions are different along the at-sea tracks of each group. The advantage of this approach is that all four groups are considered together in the same model (multi-class classification; we explain this further in response to Reviewer 3's request for more information about the random forest approach). Another analysis relating ARS to environmental conditions (presumably separately for each group) is not in line with the stated aims of our current manuscript, since it would deal more descriptively with the relationship between movement behaviour and environmental conditions for each group, rather than discerning the habitat *differences* among groups, which is our aim here.

L177-179: Moreover, one thing that I noticed is that you used all the positions to generate utilisation distributions and do not filtered the positions where birds engage in ARS or searching behaviour. Do you considered filtering the positions based on speed thresholds or to use clustering analysis (e.g. EMBC - Expectation-Maximization Binary Clustering) or Hidden Markov Models to select the positions indicative of searching behaviour and measure the kernel overlap over these positions only? - I really think that this study would benefit from investigating the habitat use by measuring the spatial overlap only in the foraging locations where birds initiate searching behaviour.

RESPONSE: During our revisions we explored using EMbC (Garriga et al. 2016, DOI: 10.1371/journal.pone.0151984) to classify the behaviour at locations along the track. EMbC relies on clustering locations according to speed and turning angle. We found that the resulting four behavioural modes classification did not deal well with the different movement modes present in our data (for example, very restricted movement of males versus extensive movement of females and some males). Specifically, we found that the clustering on turning angle appeared to be the less informative part of the classification. In our experience, this was illustrative of the difficulties of detecting behaviour modes in seabird tracking data that is common across many methods. However, to decrease any bias that may result from including locations where birds

were not foraging, we calculated a speed threshold (by calculating Fisher-Jenks breaks on the speed values), and excluded locations where speed was greater than 7.2 m/s. This threshold was more conservative than that suggested by EMbC analysis. The rationale is that high-speed movements are unlikely to represent locations at which birds are foraging. We have modified the text thus:

“Before calculating utilisation distributions, we excluded locations where the preceding displacement required a speed >7.2m/s, with the rationale that these locations were unlikely to represent foraging and would thus bias the utilisation distributions. The speed threshold was calculated using the Fisher-Jenks algorithm in the *classInt* package (Bivand 2020).” [L190-193]

Reviewer: 2

This manuscript provides a good description of how northern and southern giant petrels partition trophic resources and spatially at Marion Island. The authors present three years of tracking data and associated stable isotope values from plasma (short-term concurrent time period). They find that females of the two species have distinct distributions at sea, while the males forage closer to shore or on land. All four categories had low overlap in their isotopic niches, but when just considering birds that made at-sea distant trips male and female NGPs isotopic niches overlap. I have numerous questions about interpretation, structure, and clarity, but I found the study informative.

Main comments

An indication of how “good” your study years would be helpful in informing the discussion. Potentially this could be a reproductive success, breeding number, or a metric of carcass numbers. While not perfect this could help provide context for how abundant prey was, the degree of competition, and how pressing the need might be to exploit other resources – this seems most relevant for the males.

RESPONSE: See next response.

Annual differences or similarities are not addressed. As annual variation is often a strong driver of seabird diets and distributions please provide some justification for why you decided to pool the years.

RESPONSE:

Our exploratory analyses indicated no significant differences in space-use between years. Nonetheless, in our reanalysis we constrained permutations by year to minimise the potential effect of any interannual effects on spatial distribution (that is, labels could only be swapped within years). This did not qualitatively alter our results (and made only very minor quantitative differences).

“Since interannual variation in resource availability and environmental conditions may influence foraging behaviour, we constrained the permutations by year”. [L198-199]

In terms of how 'good' each year was, the number of breeding pairs per year was not available to us. Unfortunately, we do not have good information on prey availability each year either. The most reliable information – annual number of southern elephant seal pups born at the island – did not vary significantly among the study years (552, 557 and 623 in 2015-2017, respectively; University of Pretoria, unpublished data). However, this is not an ideal proxy since the number of carcasses should be counted to better assess food availability ashore. In the revised Discussion we mention the potential role of interannual variation in carcass availability, and in our conclusion we stress that interannual variation should be investigated further:

“Future work should also address the role of interannual variation in resource availability on segregation patterns, particularly in males ashore.” [L434-436]

However, we do note that an earlier study at Marion Island found no clear relationship between land-based prey availability (the numbers of various seal and penguin species) and breeding parameters of giant petrels (De Bruyn et al. 2007). It may be that birds compensate for resource variability through their foraging behaviour such that the variability is not observed in their breeding parameters, or that minor changes in resource availability do not observably influence foraging behaviour. That study concludes:

“Our data suggest no appreciable overlap whereby the same land-based resource singularly regulates either giant petrel’s population. A complex suite of factors related to diet and foraging (both marine and land-based), and other factors, such as the availability of appropriate nest sites, most likely interplay to regulate populations of these sibling species.”

De Bruyn PJN, Cooper J, Bester MN, Tosh CA (2007) The importance of land-based prey for sympatrically breeding giant petrels at sub-Antarctic Marion Island. *Antarct Sci* 19:25–30

In the introduction the following hypothesis is given “Given the species’ recent divergence and marked sexual dimorphism, we predict that segregation between sexes should be greater than that between species, but that the two species should also show some segregation along one or more niche-axes, which has developed in the 0.2 million years since their divergence.” This seems to generally hold true, but the very distinct geographic distributions of the females of the two species seem more distinct than the male-female segregation particularly for the NGPs. Returning to this in the discussion would be helpful.

RESPONSE: We return to this issue in our revised conclusion, tying the results back to a potential eco-evolutionary scenario:

“The intraspecific segregation between males and females, which may be driven by competitive exclusion of the smaller females from carrion resource ashore, is more marked than interspecific segregation. However, the geographic segregation of northern and southern giant petrels at sea is striking. As suggested by Hunter (Hunter 1987), the parallel pattern of sexual segregation in the two species indicates that sexual segregation arose before speciation in the giant petrels 0.2 mya (Techow et al. 2010), and the use of different geographic foraging areas at sea may represent the foraging preferences of ancestral populations that have persisted after secondary contact (Hunter 1987). These foraging preferences may have arisen in sympatry, not

necessarily due to competition, or evolved in allopatry.” [L424-432]

While great for completeness, this manuscript relies heavily on supplemental figures and tables and at times that is distracting for the reader. The stable isotope analysis section has three tables and two figures devoted to it in the main text (not supplemental). Likewise the two figures with maps in the main text seem somewhat redundant. I suggest balancing the data shown in the main manuscript and reducing the reliance on supplemental materials.

RESPONSE:

We retained Figure 1 since it maps the main feature found to differentiate northern and southern giant petrel habitats – sea surface temperature – while showing the GPS tracking locations (which are not shown in Figure 2). We replaced Figure 3 with supplementary Figure S5, which has seven in-text mentions. We deleted Supplementary Figure S4, S6 and S7. There are now only two supplementary tables (both too large for the main text) and three supplementary figures (which we deem less important than those in the text).

In line Comments:

L50 prey type, size, or?

RESPONSE: Prey type and size. Changed to “Sympatric jaguar (*Panthera onca*) and puma (*Puma concolor*), for example, had highly overlapping space and time-use, but differed in the prey type and size they selected”. [L49-51]

L51 remove internal parenthesis – issue throughout the paper

RESPONSE: We have now formatted the references as required by RSOS. Note that the responses here use the original full format, for readability.

L58 “structures”

RESPONSE: Deleted in revised text.

L45-59 The examples in this paragraph are interesting, but they feel like a list of disparate examples of the many permutations of differences in diet/space/time between similar species. A better description of the linking concepts is needed, particularly to understand why the authors chose to highlight these examples. The first sentence and the concluding sentence of the paragraph seem to be expressing the same concept.

RESPONSE: We have modified the paragraph to clarify that resource segregation typically occurs along three axes (diet, space and time) and that interacting species may only segregate along one axis, while overlapping on others (thus clarifying why

we considered these particular ‘permutations’, although we tried to use taxonomically diverse examples) – this is important for our paper, since we consider resource use along the axes introduced in these examples. We now also clearly state which of these segregation axes each example illustrates and we have removed one example (fine scale resource partitioning) to make the examples less disparate. We note that Reviewer 1 requested seabird examples in the introduction – we added references elsewhere, but here we replaced the allochryony example with a seabird one and stated as a segue at the beginning of paragraph three that inter- and intra-specific segregation is frequently described in seabirds. We deleted the last sentence (but added the reference to the first paragraph) as it does indeed paraphrase the first sentence. The revised text reads:

“It follows that resource use must be partitioned in some manner to alleviate similarity, and this segregation typically occurs along three axes: diet, space (habitat) or time (Pianka 1969, Schoener 1974). Competitors may segregate along one axis while overlapping along others. Sympatric jaguar (*Panthera onca*) and puma (*Puma concolor*), for example, had highly overlapping space and time-use, but segregated along the dietary axis, differing in the prey type and size they selected (Scognamiglio et al. 2003). Alternatively, species may mitigate resource overlap by segregating habitat-use. For example, a sibling pair of Rhinolophus bat species used different foraging habitat types when they occurred in sympatry, but this habitat preference disappeared in allopatry (Salsamendi et al. 2012). Even when species use the same resources and the same habitats, segregation along the temporal axis can allow species to exploit similar resources without direct competition. For example, a slight difference in the phenology of Adélie (*Pygoscelis adeliae*) and chinstrap (*P. antarcticus*) penguins means that the two species substantially reduced spatial overlap by foraging in similar areas a few weeks apart (Clewlow et al. 2019)”. [L47-58]

L61-68 Are we choosing between the ‘forage selection hypothesis’ and the ‘activity budget hypothesis’? “Among the explanations” implies that there are other explanations not discussed here. This paragraph needs editing to clearly highlight how males and females may or may not partition resources.

RESPONSE: We have modified this section, clarifying that there are two main hypotheses for birds:

“At the same time, males and females of the same species might be segregated on these three axes, such that they have different diets (e.g., Beck et al. 2007), use different habitats (e.g., Cleasby et al. 2015) or time their activities differently (e.g., Marcelli et al. 2003). Two main hypotheses aim to explain these patterns in birds (Catry & Phillips 2005). The ‘social dominance hypothesis’ posits that segregation results from the exclusion of subordinates by dominant conspecifics (Catry & Phillips 2005). The ‘specialisation hypothesis’ suggests that males and females segregate habitats and/or diets due to different requirements, opportunities and constraints related to their morphology, physiology and reproductive roles (Catry & Phillips 2005, Wearmouth & Sims 2008)”. [L60-66]

L70-86 These seem like one paragraph rather than two.

RESPONSE: We have made this a single paragraph.

L112: given the large body of work on this topic, “for the first time” seems like an overstatement and is not a strong or informative justification of this study.

RESPONSE: We deleted ‘for the first time’.

L119: reference needed for “0.2 million years...”

RESPONSE: We have added the reference (used at the first mention of this divergence time, earlier in the text) here too (Techow et al. 2010).

L123-24: It would be useful to introduce stable isotopes earlier in the introduction.

RESPONSE:

We moved parts of this section to an earlier, slightly expanded paragraph that introduces tracking and stable isotope analysis as two complementary sources of information to study segregation. The new paragraph reads:

“Niche segregation among related species, and between the sexes, has often been found in seabirds (e.g., Cherel et al. 2008, Navarro et al. 2009, Young et al. 2010, Bodey et al. 2014, Paiva et al. 2017, Jones et al. 2020) and two complementary tools have been particularly useful in providing this information. Tracking data allows us to quantify the space use of seabirds and, in conjunction with remote sensing data, allows us to quantify the environmental conditions – and thus the habitat – utilised by them (Wakefield et al. 2009). Measurements of $\delta^{13}\text{C}$ and $\delta^{15}\text{N}$ stable isotope values in tissues such as blood and feathers are reliable biomarkers of foraging habitat and trophic level, respectively (Inger & Bearhop 2008). These complementary sources of information capture both scenopoetic (biophysical condition or setting) and bionomic (resource) axes of the species’ niches (*sensu* Hutchinson 1978).” [L70-77]

The reduced text at the end of the Introduction reads:

“To test this hypothesis, we use high resolution tracks from animal-borne GPS devices, and the carbon and nitrogen stable isotope ratios ($\delta^{13}\text{C}$ and $\delta^{15}\text{N}$) in the blood plasma of the tracked, and some untracked, individuals. Integrating these data types for sympatric-breeding *Macronectes* populations, we show segregation in their habitat use and foraging ecology that may have evolved as a competition avoidance mechanism”. [L121-125]

L126: Given the number of tracking/stable isotope studies of just seabirds, “novel” seems out of place here.

RESPONSE: We referred in this case specifically to such data integration for sympatric breeding giant petrels, but we have deleted ‘novel’.

L138: Since your study site is on Marion Island it would be useful to know how many of these breeding pairs are on that island.

RESPONSE: We have added information for Marion Island specifically, and added more recent estimates as well as trend information, as requested by Reviewer 1.

“In 2013/2014, Marion Island had about 443 breeding pairs of NGPs (increasing since the early 2000s) and about 1,583 breeding pairs of SGPs (stable since the late 1990s) (Ryan et al. 2009, ACAP 2020). Together with Prince Edward Island, this represents around 6% of their respective global populations (Ryan et al. 2009)”. [L137-140]

L143: This implies that if they had a nesting location and phenology similar to NGP, but a bill tip of SGP they would be classed as NGP?

RESPONSE: Nesting location and phenology informed the initial species assignment, but the final species assignment was based on bill colouration. We have modified the statement as such:

“Species were differentiated by coloration of the bill tip (Hunter 1987) with nesting site and breeding phenology providing additional context for species assignment (Cooper et al. 2001).” [L144-145]

L148: Handling times would be a useful thing to report.

RESPONSE: We have added the handling times:

“Handling time was typically 5-10 minutes per bird”. [L152]

L156: Why/how did you chose the other 22 birds for molecular sexing?

RESPONSE: The other 22 birds were chosen randomly (roughly equal numbers from each species) to confirm the putative sex assignment from bill morphology. We have modified the section to clarify this:

“For four birds with no bill measurements, sex was determined genetically using the primers 2550F and 2718R (adapted from Fridolfsson & Ellegren 1999). We additionally determined the sex of 22 other birds genetically to confirm that sex assignments based on bill measurements were generally valid.” [L158-161]

L170: Will you archive this and create a doi for the repository? Thank you for including your codes!

RESPONSE: We are glad you found the repository useful. Yes, we will archive the Github repository and generate a DOI using Zenodo.

L168: The sampling interval of the GPS tags should be reported at the beginning of this section.

RESPONSE: Added:

“GPS loggers recorded locations at ~ 60-minute intervals”. [L174-175]

L172: Did this filtering remove locations?

RESPONSE: Filtering removed 9 out of 39,873 locations. We have added this information:

“After filtering (which removed ~0.02% of locations) and visual inspection, the tracks of 94 individuals (49 NGPs and 45 SGPs) were retained for further analyses.” [L178-180]

L178: Units for h?

RESPONSE: We have added the units for h (degrees). In response to Reviewer 1, we also reanalysed kernel utilisation distributions, excluding locations above a certain speed threshold and using updated h-values.

L182-185: This method has been used before. It would be good to include a citation or two.

RESPONSE: We have added a recent example where the method has been used:

Jones CW, Phillips RA, Grecian WJ, Ryan PG (2020) Ecological segregation of two superabundant, morphologically similar, sister seabird taxa breeding in sympatry. Mar Biol 167:45

L187: You need to include general details of the seven environmental data chosen. You could simply say ‘sea surface temperature’ and then use the supplementary table to provide the source of the data. Move the list of variables in L196-197 to the top of the paragraph.

RESPONSE: We moved the list to the top of the paragraph:

“These were: sea surface temperature, chlorophyll-a concentration, ocean depth, sea surface height anomaly, meridional wind velocity, zonal wind velocity and eddy kinetic energy (details in Supplementary Table S2)”. [L204-206]

L187: I guess these variables are not for the locations on land and just for the at-sea

distributions? Specify “at-sea habitat.” Potentially incorporating land variables would be a useful approach for the male giant petrels.

RESPONSE: Yes, these are mainly variables for at-sea locations and we now specify this: “To characterise the at-sea habitat used by individuals...’ [L202]. It would certainly be interesting to examine the terrestrial covariates driving the habitat use of males ashore, but we believe that is beyond the scope of this manuscript. While covariates for at-sea habitat use are well-known, this is not the case for seabirds ashore and we feel that developing and testing a set of terrestrial covariates to model the habitat use of giant petrel males ashore is something that would need to be tackled as an independent paper, since it would require very different environmental information and a separate set of analyses. The land and sea variables would require fitting two separate models, since each set of covariates would be mutually excluded from a single random forest model. Moreover, our ecological interpretation is that the habitat use of males ashore is driven mainly by the location of seal and penguin colonies.

L206-208: It is unclear which values you report.

RESPONSE: We already state clearly on L258-259 that we report the non-lipid-extracted $\delta^{15}\text{N}$ values and the lipid-extracted $\delta^{13}\text{C}$ values:

“Hereafter, we use $\delta^{15}\text{N}$ values from the raw plasma samples and $\delta^{13}\text{C}$ values from the lipid-extracted plasma samples.” [L239-240].

L225: Does this mean your sample size was 90 or less than 90?

RESPONSE: We have adjusted the wording to make it clear that the sample size was 90:

“For further analyses, we used $\delta^{13}\text{C}$ and $\delta^{15}\text{N}$ values for the 90 birds where there was enough plasma for lipid-extraction”. [L242-243]

L227: Moving (Mardia 1970) to just before the comma would make this sentence easier to read.

RESPONSE: We have moved the reference as suggested:

“We assessed multivariate normality with Mardia’s skewness and kurtosis coefficients (Mardia 1970), tested in the *MVN* package (Korkmaz et al. 2014)”. [L243-244]

L236: Since some individuals clearly foraged locally (at a known latitude), how does this correction factor influence your results? This needs to be explained and justified. You could be correcting for a combined penguin, seal, burrowing petrel signal?

RESPONSE: This correction uses the $\delta^{13}\text{C}$ value to standardise the $\delta^{15}\text{N}$ value – it corrects trophic level for the latitudinal $\delta^{13}\text{C}$ effect in the *provenance of the prey* (not

necessarily the foraging location of the individual). The $\delta^{13}\text{C}$ and $\delta^{15}\text{N}$ values in the predator both reflect the combination of prey eaten and as such the same bias will be present in both values. This is true for individuals that foraged at sea too, where the predator $\delta^{13}\text{C}$ and $\delta^{15}\text{N}$ values may reflect, for example, a combination of squid and fish. The same principle applies across any dietary inference using stable isotope analyses for individuals with mixed diets. Since we do not use the Studentised $\delta^{13}\text{C}$ values for any inference around foraging location, we believe it is unnecessary to explain and justify this further in the text. We have made a small adjustment to the text:

“While $\delta^{15}\text{N}$ values are mainly used to indicate the trophic position of animals, they are influenced by differences in baseline $\delta^{15}\text{N}$ values that reflect the isotopic gradients in the Southern Ocean (e.g., (Jaeger et al. 2010, Carpenter-Kling et al. 2020)). These gradients are captured mainly in the $\delta^{13}\text{C}$ values, but when comparing $\delta^{15}\text{N}$ values originating from different ecosystems, differences in baseline $\delta^{15}\text{N}$ values need to be considered. To account for this effect when looking at trophic position, we fitted a linear regression of $\delta^{15}\text{N}$ values against $\delta^{13}\text{C}$ values and calculated the Studentised residuals from this regression (the $\delta^{15}\text{N}$ Studentised residuals) (Bearhop et al. 2006), giving us the relative trophic positions of individuals while controlling as far as possible for the geographic source of their diet (the $\delta^{13}\text{C}$ values)”. [L252-259]

On this topic, we note that in the stable isotope biplots we only show latitudinal information (location of fronts) for individuals that foraged at sea, where we can assume that the prey $\delta^{13}\text{C}$ values are more representative of the in-situ baseline $\delta^{13}\text{C}$ values; we don't show this information in the biplot for all individuals.

L246: It seems a bit backwards to report your guess for the sexes and then confirm them with molecular sexing. Why not say first that your molecular sexing confirmed your grouping and then report?

RESPONSE: We have reversed the two pieces of information as suggested:

“Molecular sexing confirmed all putative sex assignments based on bill measurements. Culmen length of putative males ranged from 98.2 – 110.3 mm, and from 87.9 – 95.7 mm in putative females. There was no clear differentiation in bill depth, which ranged from 27.8 – 45.0 mm. Neither culmen length ($t_{84.15} = 0.628$, $p = 0.532$) nor bill depth ($t_{83.33} = -0.182$, $p = 0.856$) was significantly different between species, indicating greater intersexual size differences than interspecific differences (Supplementary Figure S2).” [L263-268]

L252: You should provide some indication of how long birds were tracked for. 1 day, 5days, 2 weeks??

RESPONSE:

Tracking durations are plotted in Supplementary Figure S1, but we have added summary statistics in the main text, also differentiating the logger attachment duration from the track duration after excluding on-nest locations:

“Birds were tracked for 7.8 – 31.0 days (mean = 16.6 days). After trimming the tracks to exclude locations on the nest, tracks were 1.4 – 24.3 days long (mean = 9.1 days) (Supplementary Figure S1).” [L271-273]

L253: You don't provide any quantitative methods for how you determined these two foraging strategies. These need to be included. Why do the females fit into this group if they are the outliers? It seems like ~90km is too far to be foraging on penguin/seal resources or in-shore waters around Marion?

RESPONSE: We initially delineated these two foraging strategies by looking at plots of the distribution of maximum foraging distance (histograms and ordered value curves). We have re-examined this delineation using two quantities methods:

1) k-means clustering (with $k = 2$) by trip duration and trip distance. This yields a break at 729 km.

2) Calculating univariate breaks on the trip distance alone. The Jenks method (729 km) and univariate k-means (788 km) are similar to bivariate k-means above (729 km), but quantile breaks yields a division at 55 km.

These two results show that an ecologically-informed visual classification can be more appropriate than a naïve quantitative approach. We recognize that 90 km is a generous threshold for distant trips, as the reviewer points out, and in the revised version we therefore adopt the quantile breaks threshold (55 km). This did not qualitatively affect the results. We added the following text to the manuscript:

“We quantitatively distinguished these two strategies by calculating quantile breaks on trip distance using the *classInt* package (Bivand et al. 2020). We tested other class classification methods, and bivariate k-means, but these results were not ecologically sensible.” [L180-183]

L253: Were birds that were categorized in the second group tracked for longer? I.e. Long enough to make both the short trips observed in the first group AND a long trip?

RESPONSE: Individuals that made long trips were tracked slightly, but significantly, longer (mean for birds that made long trips = 17.7 days, mean for birds that did not make long trips = 13.2 days, $t_{56.15} = 4.33$, $p < 0.001$). However, this is an artefact of a fieldwork constraint. Approximately 3 days after deployment we began checking whether tracked birds were back at the nest for logger retrieval. Since long trips take longer than short ones, birds making long trips were likely to arrive back at the nest later than birds that did not (thus causing the observed difference). However, if we look at the typical duration of long trips (0.2 – 18.0 days, mean = 7.3 days) in light of the overall tracking duration (figure below), we see that birds that did not make long trips (left boxplot) had more than enough time to do so (mean deployment duration on those birds that did not make long trips = 13.2 days, range = 7.8 – 21.0 days). Thus, we are confident that the two observed foraging strategies are not merely an outcome of different tracking durations.

L281: Figure 3 is hard to interpret since the points are plotted on top each other.

RESPONSE: We replaced this figure with Supplementary Figure S5.

L288: It isn't surprising that at the 50% Utilization distribution scale that the males wouldn't be segregated. Without knowing the unit of the smoothing factor h it is hard to know how "smoothed" these distributions are relative to the data, but from your point data it seems like they might be very smoothed. Did you see finer scale clustering and separation between the males in the GPS data or was your GPS data too low temporal resolution to tell?

RESPONSE: We have updated the utilization distribution analysis in two ways: 1) we applied a velocity threshold prior to calculating the utilization distributions, thereby excluding putative transit locations (see response to Reviewer 1); 2) we recalculated the utilization distributions with updated h -values, including a much lower h -value for males that made only short trips (see Figure 2b). Even with these changes, males were not significantly segregated. We interpret this as males of both species foraging in the same penguin and seal colonies ashore.

L328: I am not sure that you explicitly demonstrate this with your data.

RESPONSE: We deleted 'the same' and replaced 'habitats' with 'areas' since it is true that we do not explicitly (using the random forest model) demonstrate that males use the same habitat. However, we do clearly demonstrate spatial overlap (Figure 1b, 2b, Table 1) and have therefore left the rest of the sentence the same:

"Males of both species tended to stay close to the island where they were not spatially segregated, exploiting foraging areas on land near their nests." [L342-343]

L329/L332 The statements “exploiting the same resources” and “males were isotopically segregated” seem contradictory.

RESPONSE: The statement on L329 very clearly refers to males that made at-sea foraging trips, and they behaved in a similar way to females, while L332 compares males of the two species. We have clarified thus:

“Males of the two species exploited resources at a similar trophic level ($\delta^{15}\text{N}$ Studentised residuals), but with different origins, evidenced by slight differences in $\delta^{13}\text{C}$ values. Thus, males of the two species were isotopically segregated (along the $\delta^{13}\text{C}$ axis) despite using the same foraging habitats”. [L345-348]

L338 replace “utilized” with “used”

RESPONSE: Replaced.

L348-351: You included oceanographic habitat analysis in this paper. How do your results support this statement?

RESPONSE:

We added: “The specific situation of Marion Island near the Antarctic, Sub-Antarctic and Subtropical waters, allows giant petrels at Marion Island to exploit these very different habitats and corresponding different prey types.” [L366-368]

L354 Integrating the observations of feeding with the first paragraph would be useful.

RESPONSE: See response below about how we have integrated information across the two paragraphs.

L359 Is this because the SGP go further south or are just foraging on other species that range further south? Since you are looking at plasma it doesn't seem like the first would fit with your tracking data results. Clarify and integrate the tracking into this discussion.

RESPONSE: As we already state in the manuscript, the integration time of blood plasma fits well with the tracking periods and thus the GPS tracking data should reflect the same period as the plasma stable isotope ratio values, as you note. This, in conjunction with our observed $\delta^{15}\text{N}$ values, supports the idea that SGPs may feed on slightly different prey, even on land. We have rearranged the paragraph to clarify:

“While the trophic level of males that foraged on or near land was similar between species, the slightly lower $\delta^{13}\text{C}$ values of SGP males suggests that they use a different prey resource on land, with a more southerly origin (since $\delta^{13}\text{C}$ baseline values in the Southern Ocean decrease at more southerly latitudes (Trull & Armand 2001), leading to subtle isotopic segregation. If males frequently alternate their foraging strategy between scavenging on land and pelagic trips, this might influence their $\delta^{13}\text{C}$ values, explaining the small difference, but this is not evident in the $\delta^{15}\text{N}$ values or supported by the short integration time of blood plasma, which corresponds well with the tracking periods. The 5-6 week difference in breeding timing between the

species has been suggested to be an important segregation mechanism between the two species in general (Hunter 1984, Cooper et al. 2001, Granroth-Wilding & Phillips 2019) but it does not free males from broad spatial overlap, as we show with our data collected near-simultaneously for both species.” [L379-388]

L364: I think your methods may only resolve “large scale spatial overlap”, ie. you show that they are both foraging on the same island. In Figure 1 the NGPs that made short trips seem to have a much more extensive at sea distribution than the SGPs. They also seem to visit different beaches that the SGPs (that were tracked) didn’t visit.

RESPONSE: In the revised version we have calculated finer-scale utilisation distribution overlap for males (see response above). Nonetheless, we added ‘broad’:

“The 5-6 week difference in breeding timing between the species has been suggested to be an important segregation mechanism between the two species in general (Hunter 1984, Cooper et al. 2001, Granroth-Wilding & Phillips 2019) but it does not free males from broad spatial overlap, as we show with our data collected near-simultaneously for both species.” [L385-388]

L365: More details (and analysis?) are needed to explain how the timing of spatial overlap is important in order to support these statements.

RESPONSE: Since our analysis deals with near-simultaneous tracking data for the two species, the section on allochry is speculative and based on the previously-shown importance of allochry at South Georgia for giant petrels generally (Granroth-Wilding & Phillips 2019). Using the near-simultaneous tracking data we clearly demonstrate high spatial overlap among males. We have thus restricted the paragraph to remove this speculation.

“The 5-6 week difference in breeding timing between the species has been suggested to be an important segregation mechanism between the two species in general (Hunter 1984, Cooper et al. 2001, Granroth-Wilding & Phillips 2019) but it does not free males from broad spatial overlap, as we show with our data collected near-simultaneously for both species.” [L385-388]

L373-383 Since you can’t directly address this, shortening this explanation and integrating it into the prior paragraph would be helpful. Do you know if there were ample carcasses available during your study years?

RESPONSE: Unfortunately (as explained earlier), we do not have good information on the prey availability in each year, although the only measure we do have indicates no difference among years. We have integrated the information across the two paragraphs in this subsection and shortened the first one (additional information was requested in the comment above, for the second paragraph). Since we do not have the required information now, we have stated that the issue of year to year variation in resource availability should be investigated with future work. The first paragraph now reads:

“Males, in contrast to females, typically foraged on land or inshore near their breeding sites, resulting in significant sexual segregation, but high interspecific habitat overlap. Aggressive competition for carrion is commonly seen between male SGPs and NGPs and it is possible that the dominance of some males excludes others from carrion resources, forcing them to forage at sea or at lower quality sites on land (see (De Bruyn & Cooper 2005). However, culmen lengths (a proxy of body size) of males that made long trips were not different from those of males that did not make long trips. Future work could investigate resource-use patterns in males, with respect to year-to-year variation in carrion availability.” [L371-377]

We also added to the conclusion:

“Future work should also address the role of interannual variation in resource availability on segregation patterns, particularly in males ashore.” [L434-436]

L385 You Supplemental Figure 7 shows that you are missing the prey with the elevated N. Without options (and data) for your high N prey items your prey paragraph feels very speculative. I am skeptical about burrowing petrels being a high component of the diet of male giant petrels – do you have any observations from your study years?

RESPONSE: It is true that the potential prey isospace is not fully captured by the data showing in Supplementary Figure S7. Unfortunately, we do not have predation observations in our study years, although we do note here an observation of a northern giant petrel feeding on a Salvin’s prion (Jones et al. 2019, DOI: 10.1017/S0954102019000415). We have thus shortened the paragraph and reworded the text to be less speculative.

“This suggests that during the tracking period the diet of males was dominated by species feeding at a relatively low trophic level. Rather than feeding on seals, which are mainly piscivorous at Marion Island (Reisinger, Landman, et al. 2018), the low $\delta^{15}\text{N}$ values of male giant petrels suggest they feed on crustacean-feeders (see Connan et al. 2019, Jones et al. 2019). However, a better isotopic characterisation of the potential prey field is required to resolve the potential diet composition of giant petrels at Marion Island”. [L394-398].

L405-407 The Gonzalis-Solis and Granroth-Wilding study were both from Bird Island, South Georgia so it seems odd to contrast them to each other like they were from different places. It seems likely that annual variation plays a role as well. Did your study find any variation between the three years?

RESPONSE: We did not see any annual variation in our exploratory analyses (which is why we pooled the years), but in our revised analyses we have constrained the space-use and isotopic overlap permutations by year to alleviate any interannual effects. Nonetheless, interannual effects may certainly be important and in the discussion and conclusion we have recommended this as a future line of enquiry, in the context of year to year variation in prey availability. The contrast between different studies from South Georgia is indeed interesting, but it could also be due to different analytical methods (each study used only tracking or stable isotope data) and interpretation.

Our comparison with results from other studies is also subjective to some extent, of course, since consistency was not formally measured. We have clarified our comparisons to make the situation clearer while stressing the possibility of interannual variation:

“This flexibility in foraging strategies has been observed in NGPs and SGPs at South Georgia, where both males and females showed plastic foraging behaviour (Granroth-Wilding & Phillips 2019) although females at South Georgia (Forero et al. 2005, Granroth-Wilding & Phillips 2019) and Patagonia (Forero et al. 2005) displayed more foraging flexibility than we observed. Our results were more similar to the consistency reported for females at South Georgia in a different study (González-Solís et al. 2008) highlighting the possibility of significant interannual variation and the utility of integrating stable isotope and tracking data.” [L403-408].

L410: Both males and females forage at sea during the winter, right? Perhaps males exclude females from easily accessed carrion resources, but are able to supplement with at-sea resources as needed.

RESPONSE: That is correct, yes. We have slightly reworded the existing text here to make that point clearer:

“Females may be competitively excluded from carrion resources ashore and forced more often to forage at sea due to their smaller body size, while males can supplement their on-land carrion foraging with at-sea foraging when necessary. In winter, for example, studies elsewhere show that both sexes forage at sea, when carrion availability on land is low (González-Solís et al. 2008, Krüger et al. 2018, Granroth-Wilding & Phillips 2019).” [L410-414]

L419 The reference at the end of this sentence seems out of place. Perhaps a review would make more sense.

RESPONSE: This paragraph has been omitted, in response to the comment below.

L417-432 This paragraph is unexpected and out of place given how the study was framed and the hypothesis presented in the introduction. Omit or reframe to include anthropogenic threats in the introduction.

RESPONSE: We have deleted this paragraph and slightly adjusted part of our conclusion to flag this question for further investigation (since Reviewer 3 did highlight the potential species conservation implications in their general comments):

“Variation in foraging patterns among giant petrels breeding at Marion Island exposes species and sexes to different threats, which should be investigated in more detail (e.g., Thiers et al. 2014, Gianuca et al. 2019) and potentially considered in their conservation and management.” [L432-434]

L435 While it may be true “for the first time” doesn’t really speak to the interesting or informative parts of this study. Omit.

RESPONSE: Deleted ‘for this first time’.

Reviewer: 3

studying intra and interspecific segregation in seabirds is key to understand theoretical aspects of coexistence in sympatric sibling species but also in terms of species conservation.

This study investigated segregation in two sympatric sibling species using tracking, environmental and stable isotope data. The study is well executed and well written.

I suggest only minor changes to clarify certain aspects.

1. in the conclusion the authors suggest very nicely how this intra and inter specific segregation arose in evolutionary times. however, at the beginning and within the introduction they focused on the common definition of competition to address segregation, which implies limited resources which is not prove neither imply in this study. I suggest adding other theories for segregation within the introduction that may be influencing in the pattern found in the study spp (i.e. life history theory).

RESPONSE: This was also mentioned by Reviewer 1. We have rephrased the section on explanations for sexual segregation, which should make it clearer that sexual segregation is not necessarily a competition avoidance mechanism but may rather be due to life history constraints:

“The ‘specialisation hypothesis’ suggests that males and females segregate habitats and/or diets due to different requirements, opportunities and constraints related to their morphology, physiology and reproductive roles (Catry & Phillips 2005, Wearmouth & Sims 2008).” [L64-66]

We have also added to the introduction a brief statement on the other mechanisms that may lead to currently-observed niche segregation:

“Competition (whether current or in species’ evolutionary history) is frequently invoked as an explanation for niche segregation, but segregation may result from other factors including non-competitive evolutionary adaptation and stochastic processes (Begon et al. 2006)”. [L66-68]

2. I found very interesting exploring habitat use using random forest analysis, I suggest audience may be benefit if further details are given for the analyses. Also, more details in the results section regarding this will be useful.

RESPONSE: We added more details in the methods section:

“We then related these variables to species and sex using a random forest classification model – a fast and accurate method that can classify multiple target classes (Breiman 2001). The fitted random forest model predicts, for each at-sea GPS location, to which group (NGP or SGP, male or female) the location belongs based on the value of the environmental covariates. We fitted the model in the *randomForest* R

package (Liaw & Wiener 2002), growing 1000 trees. We assessed variable importance as the mean decrease in Gini index”. [L208-213]

The results already report information typically presented for random forests – the model accuracy and the rank importance of explanatory variables (we added this in the text, since we deleted Supplementary Figure S6). We note that all the scripts for the analyses are in the associated Github repository (<https://github.com/ryanreisinger/giantPetrels/>), so full details of the model fitting can be seen there.

3. Lines 349-351 what do the authors mean this this statement? How is the interpretation of oceanography influencing the foraging strategy? diet preference could influence foraging strategy and thus oceanography as prey rely on certain aspects of the ocean, please clarify.

RESPONSE: This was also mentioned by Reviewer 2. We have expanded on the statement as follows:

“The specific situation of Marion Island near the Antarctic, Sub-Antarctic and Subtropical waters, allows giant petrels at Marion Island to exploit these very different habitats and corresponding different prey types.” [L366-368]

Appendix B

14 August 2020

The handling editor,

We thank you and the referees for the comments on our revised manuscript. We have made the changes suggested by reviewer 2 and addressed the journal's manuscript preparation points. The changes are indicated in the track changes Word document accompanying the resubmission.

Sincerely,

Ryan Reisinger & co-authors

Reviewer: 2

Comments to the Author(s)

I found the manuscript much improved. Thank you for your detailed replies to the reviewer comments.

The authors have addressed the potential for annual differences in the permutation analysis, but it is unclear how annual differences were accounted for (or not) in the at-sea habitat use analysis or the stable isotope statistical analysis. If years were not accounted for this should be clearly stated in the text. Otherwise, my comments, listed below are very minor.

RESPONSE:

For analyses other than the permutation tests we retained the year-pooling. We have stated this clearly in the revised text:

'Since interannual variation in resource availability and environmental conditions may influence foraging behaviour, we constrained the permutations by year. However, for other habitat analyses (including the random forest model described below) we pooled the data for different years.'

And

'Again, however, we pooled data from the different years for the general stable isotope analyses.'

L62 insert "same" three axes...

RESPONSE:

Replaced 'these' with 'these same'.

L68 consider commas rather than parenthesis.

RESPONSE:

Replaced the parentheses with commas.

L123 "high resolution tracks" at hourly locations this "high-resolution" statement seems like an oversight. Omit

RESPONSE:

Deleted 'high resolution'.

Replace utilize with use throughout the manuscript

RESPONSE:

We have replaced all instances of ‘utilise’ (‘utilised’) with ‘use’ (‘used’).

L126: Rephrase: The conclusion of your introduction reads like a result “we show segregation in their habitat use and foraging ecology that may have evolved as a competition avoidance mechanism. Consider omitting this last statement

RESPONSE:

We deleted the sentence.

L132 Rephrase (missing comma?) “are located on a rise in water ~5000 m deep”

RESPONSE:

We have rephrased the sentence for clarity:

‘The Prince Edward Islands (46.9° S, 37.7° E), comprising Marion Island and Prince Edward Island, are located on a shallow rise surrounded by water ~5000 m deep.’

L132 Add the word “habitat” before variability

RESPONSE:

Added ‘oceanographic’ rather than ‘habitat’:

‘There is high mesoscale oceanographic variability in the vicinity of the islands, stemming from the frequent presence of eddies spawned as the west-flowing Antarctic Circumpolar Current interacts with the South-West Indian Ridge upstream of the islands.’

L154: Handling time should be reported as a value \pm standard deviation.

RESPONSE:

Handling time was not measured in each case, rather it was estimated. We rephrased the statement thus:

‘We estimated handling time was approximately 5-10 minutes per bird’.

L184: This sentence may not be needed: “We tested other class classification methods, and bivariate k-means, but these results were not ecologically sensible.”

RESPONSE:

Deleted.

L260: Rephrase for grammar: “as far as possible”

RESPONSE:

Moved ‘as far as possible’ to the end of the sentence for less awkward phrasing:

‘To account for this effect when looking at trophic position, we fitted a linear regression of $\delta^{15}\text{N}$ values against $\delta^{13}\text{C}$ values and calculated the Studentised residuals from this regression (the $\delta^{15}\text{N}$ Studentised residuals) [75], giving us the relative trophic positions of individuals while controlling as far as possible for the geographic source of their diet (the $\delta^{13}\text{C}$ values) as far as possible.’

L320-322: Rewrite to avoid 1 sentence paragraphs.

RESPONSE:

We add this one sentence paragraph to the preceding paragraph.

L321: Reverse order here for consistency: NGPs and SGPs

RESPONSE:

We reversed the order of the first mention so that it is consistently SGPs and NGPs throughout this sentence.

L336: Reiterate that this is during incubation.

RESPONSE:

Added ‘During incubation’:

“During incubation, segregation occurred along at least two axes – isotopically distinct food resources (diet) and differential habitat use (space) – illustrating how environmental resources may be partitioned among similar animals.”